# Precise Asymptotics and Refined Regret of Variance-Aware UCB

**Yingying Fan**
University of Southern California
fanyingy@marshall.usc.edu

**Yuxuan Han**
New York University
yh6061@stern.nyu.edu

**Jinchi Lv**
University of Southern California
jinchilv@marshall.usc.edu

**Xiaocong Xu**
University of Southern California
xuxiaoco@marshall.usc.edu

**Zhengyuan Zhou**
New York University
zzhou@stern.nyu.edu

## Abstract

In this paper, we study the behavior of the Upper Confidence Bound-Variance (UCB-V) algorithm for the Multi-Armed Bandit (MAB) problems, a variant of the canonical Upper Confidence Bound (UCB) algorithm that incorporates variance estimates into its decision-making process. More precisely, we provide an asymptotic characterization of the arm-pulling rates for UCB-V, extending recent results for the canonical UCB in [21] and [23]. In an interesting contrast to the canonical UCB, our analysis reveals that the behavior of UCB-V can exhibit instability, meaning that the arm-pulling rates may not always be asymptotically deterministic. Besides the asymptotic characterization, we also provide non-asymptotic bounds for the arm-pulling rates in the high probability regime, offering insights into the regret analysis. As an application of this high probability result, we establish that UCB-V can achieve a more refined regret bound, previously unknown even for more complicate and advanced variance-aware online decision-making algorithms. A matching regret lower bound is also established, demonstrating the optimality of our result.

## 1 Introduction

The Multi-Armed Bandit (MAB) problem is a fundamental framework capturing the exploration-exploitation trade-off in sequential decision-making. Over decades, it has been rigorously studied and widely applied across fields like dynamic pricing, clinical trials, and online advertising [32, 35, 27].

In the classic $K$-armed MAB problem, a learner is faced with $K$ arms, each associated with an unknown reward distribution $P_a$ for $a \in [K]$, supported on $[0, 1]$ with mean $\mu_a$ and variance $\sigma_a^2$. At each time step $t \in [T]$, the learner selects an arm $a_t$ and observes a reward $X_t$ drawn independently from $P_{a_t}$. The learner's goal is to maximize the cumulative reward by striking an optimal balance between exploration (sampling less-known arms to improve estimates) and exploitation (selecting arms with high estimated rewards). This objective is commonly framed as a regret minimization problem, where the regret over a time horizon $T$ is defined as $\mathrm{Reg}(T) \equiv \sum_{t=1}^{T}(\mu_{a^\star} - \mu_{a_t})$ with $a^\star \equiv \arg\max_{a \in [K]} \mu_a$ the optimal arm with the highest expected reward.

39th Conference on Neural Information Processing Systems (NeurIPS 2025).

| Regime | Stability | Regret Upper Bound | Regret Lower Bound |
|---|---|---|---|
| $\sigma_1 = \mathfrak{o}(\sigma_2)$ | No* | $\widetilde{\mathcal{O}}(\sigma_2 \sqrt{T})$ | $\widetilde{\Omega}(\sigma_2 \sqrt{T})$ |
| $\sigma_1 = \Omega(\sigma_2)$ | Yes | | |
| $\sigma_2 = \mathfrak{o}(\sigma_1)$ | | $\widetilde{\mathcal{O}}((\sigma_2/\sigma_1) \cdot \sigma_2 \sqrt{T})$ | $\Omega((\sigma_2/\sigma_1) \cdot \sigma_2 \sqrt{T})$ |

Table 1: Summary of stability and regret results for UCB-V under different $\sigma_1, \sigma_2$ regimes.
*There exists an instance such that the arm pulling rate is unstable.

The minimax-optimal regret for the $K$-armed bandit problems is known to be $\Theta(\sqrt{KT})$ up to logarithmic factors. This rate is achievable by several well-established algorithms, including the Upper Confidence Bound (UCB) [5, 4], Thompson Sampling [1, 32], and Successive Elimination [13], among others. Beyond the regret minimization, increasing attention has been devoted to analyzing finer properties and large-$T$ behaviors of several typical algorithms, including the regret tail distributions [15, 16], diffusion approximations [14, 21, 2, 24], and arm-pulling rates [21, 23]. Among these works, [21] and [23] introduced the concept of stability for the canonical UCB algorithm, enabling the precise characterization of arm-pulling behaviors and facilitating statistical inference for adaptively collected data, a task traditionally considered challenging for the general bandit algorithms.

In more structured settings, sharper regret bounds are achievable, particularly when arm variances are small. For instance, if all arms have zero variance (i.e., deterministic rewards), a single pull of each arm is sufficient to identify the optimal one. This observation has motivated active research into developing variance-aware algorithms for bandit problems [5, 3, 4, 20, 28, 40, 41, 33]. Among these algorithms, the Upper Confidence Bound-Variance (UCB-V) algorithm [3, 4], detailed in Algorithm 1, adapts the classic UCB algorithm by incorporating variance estimates of each arm.

While regret minimization for variance-aware algorithms has been well-studied, their precise arm-pulling behavior remains less explored. The main challenge here is that incorporating variance information introduces a new quantity, in addition to the reward gaps $\Delta_a \equiv \mu_{a^\star} - \mu_a, a \in [K]$, that influences the arm-pulling rates. In fact, the behavior of the algorithm can *differ significantly* depending on whether variance information is incorporated or not. In Figures 1 and 2b, we compare the empirical arm-pulling rates between UCB-V and the canonical UCB in a two-armed setting. Compared to the canonical UCB, its variance-aware version exhibits significantly greater fluctuations in arm-pulling numbers as the reward gap changes, and its arm-pulling distribution is more heavy-tailed. This highlights the significant differences in the variance-aware setting and introduces additional challenges.

---

**Algorithm 1** UCB-V Algorithm

---
1: **Input:** Arm number $K$, time horizon $T$, and exploration coefficient $\rho$.
2: Pull each of the $K$ arms once in the first $K$ iterations.
3: **for** $t = K + 1$ to $T$ **do**
4:     Compute arm pulls for arm $a$ up to time $t$ by $n_{a,t} \equiv \sum_{s \in [t-1]} \mathbf{1}\{A_s = a\}, a \in [K]$.
5:     Compute the empirical means and variances

$$\bar{X}_{a,t} \equiv \frac{1}{n_{a,t}} \sum_{s \in [t-1]} \mathbf{1}\{a_s = a\} X_s, \quad \widehat{\sigma}^2_{a,t} \equiv \frac{1}{n_{a,t}} \sum_{s \in [t-1]} \mathbf{1}\{a_s = a\}(X_s - \bar{X}_{a,t})^2.$$

6:     Compute the optimistic rewards $\mathrm{UCB}(a,t) \equiv \bar{X}_{a,t} + \left(\frac{\widehat{\sigma}_{a,t}}{\sqrt{\rho \log T}} \vee \frac{1}{\sqrt{n_{a,t}}}\right) \frac{\rho \log T}{\sqrt{n_{a,t}}}$ for $a \in [K]$.
7:     Choose arm $A_t$ given by $A_t = \arg\max_{a \in [K]} \mathrm{UCB}(a,t)$.
8: **end for**

---

## 1.1 Our Contributions

In this work, we try to close this *gap* by presenting a precise asymptotic analysis of UCB-V. As in [21] and [23], our main focus is on the arm-pulling numbers and stability. For clarity, we concentrate on the two-armed bandit setting, specifically $K = 2$, in the main part of our paper, as it effectively captures the core exploration-exploitation trade-off while allowing a precise exposition of results [31, 18, 22, 21]. The extension to the $K$-armed case is discussed in Appendix B.

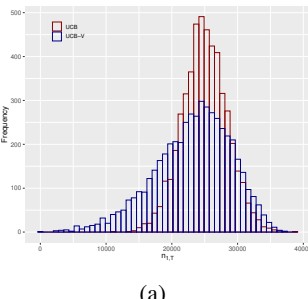 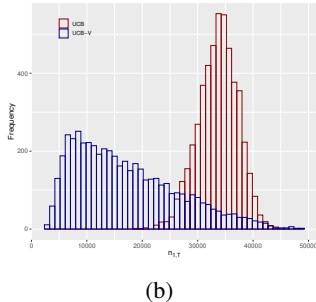

(a)                                           (b)

Figure 1: The distributions of $n_{1,T}$ for UCB-V and UCB with $T = 50,000$ over 5000 repetitions. (a) $\sigma_1 = \sigma_2 = 1/4, \Delta_2 = 0$ (b) $\sigma_1 = 0, \sigma_2 = 1/4, \Delta_2 = \sigma_2\sqrt{\log T / T}$. A more detailed numerical setting is provided in Appendix H.

More precisely, by providing a general concentration result, we establish both precise asymptotic characterizations and high-probability bounds for the arm-pulling numbers of UCB-V. When $\sigma_1, \sigma_2 = \Omega(1)$, our results provide a straightforward generalization of those obtained for the canonical UCB. In contrast, when $\sigma_1 \ll \sigma_2$, we show that, unlike UCB, the stability result may *not* hold for UCB-V and that it reveals a *phase transition* phenomenon in the optimal arm-pulling numbers, as illustrated in Figure 2. Finally, as an application of our *sharp* characterization of arm-pulling rates, we present a *refined* regret result for UCB-V that is *previously unknown* for any other variance-aware decision-making algorithms. We summarize our contributions in more detail below (see also Table 1). For notational simplicity, we assume that the optimal arm is $a^\star = 1$ and denote by $\Delta = \Delta_2$.

**Precise asymptotic behavior for UCB-V.** For fixed values of $\Delta, \sigma_1, \sigma_2$, and $T$, we introduce a pair of deterministic equations whose unique solution $\{n_{a,T}^\star \equiv n_{a,T}^\star(\Delta, \sigma_1, \sigma_2, T)\}_{a=1,2}$ predicts the arm-pulling numbers of UCB-V. We show that for possibly $T$-dependent $(\Delta, \sigma_1, \sigma_2)$,

$$\frac{n_{a,T}}{n_{a,T}^\star} \xrightarrow{p} 1 \quad \text{for } a = 1, 2,$$

except when $\sigma_1 \ll \sigma_2$ and $\sigma_2\sqrt{\rho \log T}/(\sqrt{T}\Delta) = 1$ hold simultaneously; see Theorem 1 for a precise statement. This phenomenon, referred to as the stability of arm-pulling numbers, is formally defined in Definition 1. Consequently, the solutions of these deterministic equations can be used to predict the behavior of UCB-V in the large-$T$ regime, as illustrated in Figure 2. Our results reveal several notable differences between UCB-V and the canonical UCB, as analyzed in [21] and [23], due to the incorporation of variance information:

- When $\sigma_1, \sigma_2 = \Omega(1)$, the optimal arm will always be pulled a linear number of times, with $n_{1,T} \sim \frac{\sigma_1^2}{\sigma_1^2 + \sigma_2^2}T$ as $\Delta \to 0$. This indicates that, in the small-gap regime, the UCB-V algorithm tends to allocate more pulls to the arm with higher reward variance, generalizing the result for the canonical UCB in which both arms are pulled $T/2$ times in the small-$\Delta$ regime.

- When $\sigma_1 = \mathfrak{o}(\sigma_2)$, UCB-V may pull the optimal arm sublinearly in the small-$\Delta$ regime. In contrast, the canonical UCB pulls the optimal arm linearly in $T$, as illustrated in Figure 2b. This behavior is governed by the ratio $\Lambda_T \equiv \sigma_2\sqrt{\rho \log T}/(\sqrt{T}\Delta)$. More precisely, given $\sigma_1 \leq \sqrt{\rho \log T / T}$ and $\sigma_2 = \Omega(1)$,

  1. When $\overline{\lim}_{T\to\infty} \Lambda_T < 1$, we have $n_{1,T}/n_{1,T}^\star \xrightarrow{p} 1$ with $n_{1,T}^\star = \Omega(T)$.

  2. When $\underline{\lim}_{T\to\infty} \Lambda_T > 1$, we have $n_{1,T}/n_{1,T}^\star \xrightarrow{p} 1$ with $n_{1,T}^\star = \mathcal{O}(\sqrt{T}/\sigma_2)$.

  3. When $\lim_{T\to\infty} \Lambda_T = 1$, there exists a bandit instance where, for large $T$, $\mathbb{P}(n_{1,T} \gtrsim T/\sqrt{\log T}) \wedge \mathbb{P}(n_{1,T} \lesssim \sqrt{T}\log T) \gtrsim 1$.

  The above results completely characterize the behavior of $n_{1,T}$ in the prescribed regime. Notably, we establish a phase transition in the optimal-arm pulling times $n_{1,T}$, shifting from $\mathcal{O}(\sqrt{T}/\sigma_2)$ to $\Omega(T)$ at the critical point $\Lambda_T = 1$. We also note that the existence of an *unstable* instance when $\Lambda_T \sim 1$ highlights a *stark contrast* between UCB-V and the

canonical UCB, where, for the latter, it has been shown that the behavior of $n_{1,T}$ for any $\Delta > 0$ can be asymptotically described by a deterministic sequence $\{n_{1,T}^\star\}$ as $T \to \infty$.

**High probability bounds and confidence region for arm pulling numbers.** While our asymptotic theory provides the precise limiting behavior of UCB-V, it has a drawback similar to those found in [21] and [23]: the convergence rate in probability is quite slow. Specifically, the probability that the uncontrolled event occurs decays at a rate of $\mathcal{O}((\log T)^{-1})$. This slow rate is inadequate for providing insight or guarantees in the popular *high probability regime* for studying bandit algorithms, where the probability of an uncontrolled event should be in the order of $\mathcal{O}(T^{-1})$. To address this *gap*, we demonstrate that, starting from our unified concentration result in Proposition 1, one can derive non-asymptotic bounds for the arm pulling numbers in the high probability regime. Such result provides a high probability confidence region for arm pulling numbers, as illustrated in Figure 3a in Appendix D.1.

**Refined regret for variance-aware decision making.** As an application of our high probability arm-pulling bounds, we demonstrate in Theorem 2 that the UCB-V algorithm achieves a refined regret bound of form

$$\mathcal{O}\left(\frac{\sigma_2^2}{\sigma_1 \vee \sigma_2}\sqrt{T}\right).$$

This result improves upon the best-known regret[1] for UCB-V [4, 36] and surpasses the regrets for all known variance-aware bandit algorithms [41, 40, 33, 9, 10], which are of form $\mathcal{O}(\sigma_2\sqrt{T})$ in the two-armed setting. Our result is the *first* to reveal the effect of the optimal arm's variance: When $\sigma_1 = \mathfrak{o}(\sigma_2)$, the regret matches the previously known $\mathcal{O}(\sigma_2\sqrt{T})$ bound, and the optimal arm's variance does not affect performance; as $\sigma_1$ surpasses $\sigma_2$, the regret decreases as $\sigma_1$ increases, following the form $\mathcal{O}(\frac{\sigma_2^2}{\sigma_1}\sqrt{T})$. Simulations presented in Figure 2 confirm our theoretical predictions, demonstrating that UCB-V exhibits improved performance in high-$\sigma_1$ scenarios, which was *previously unknown*. A matching regret lower bound is established in Theorem 3, demonstrating the optimality of our result.

**Notation.** For any positive integer $n$, denote by $[n] = [1 : n]$ set $\{1, \ldots, n\}$. For $a, b \in \mathbb{R}$, $a \vee b \equiv \max\{a, b\}$ and $a \wedge b \equiv \min\{a, b\}$. For any $a \in \mathbb{R}$, $a_+ \equiv \max\{a, 0\}$. Throughout this paper, we regard $T$ as our fundamental large parameter. For any possibly $T$-dependent quantities $a \equiv a(T)$ and $b \equiv b(T)$, we say that $a = \mathfrak{o}(b)$ or $b = \omega(a)$ if $\lim_{T \to \infty} a/b = 0$. Similarly, $a = \mathcal{O}(b)$ or $b = \Omega(a)$ if $\lim_{T \to \infty} |a/b| \leq C$ for some constant $C$. If $a = \mathcal{O}(b)$ and $a = \Omega(b)$ hold simultaneously, we say $a = \Theta(b)$, or $a \asymp b$, and write $a \sim b$ in the special case when $\lim_{T \to \infty} a/b = 1$. We write $a \gtrsim b$ (resp. $a \lesssim b$) to mean $a \geq Cb$ (resp. $a \leq Cb$) for some absolute constant $C > 0$. If either sequence $a$ or $b$ is random, we say $a = \mathfrak{o}_\mathbf{P}(b)$ if $a/b \xrightarrow{p} 0$ as $T \to \infty$.

## 2 Implicit bounds for arm-pulling numbers

**Additional notations.** For $\rho > 1$ and $T \in \mathbb{N}_+$, let

$$\overline{\sigma}_a(\rho; T) \equiv \frac{\sigma_a}{\sqrt{\rho \log T}}, \qquad \overline{\Delta}_a(\rho; T) \equiv \frac{\Delta_a}{\rho \log T}, \quad a \in [K].$$

Whenever there is no confusion, we write $\overline{\sigma}_a \equiv \overline{\sigma}_a(\rho; T)$ and $\overline{\Delta}_a \equiv \overline{\Delta}_a(\rho; T)$. We also allow $\sigma_a$ and $\Delta_a$ to depend on $T$. For any $\sigma \geq 0$ and $n \in \mathbb{N}_+$, define $\varphi(n; \sigma) \equiv \sigma \vee n^{-1/2}/\sqrt{n}$. For each fixed $\sigma \geq 0$, the map $\varphi(\cdot; \sigma) : \mathbb{R}_{\geq 0} \to \mathbb{R}_{\geq 0}$ is monotone nonincreasing in $n$, and its (piecewise) inverse $n(\cdot; \sigma) : \mathbb{R}_{\geq 0} \to \mathbb{R}_{\geq 0}$ is $n(\varphi; \sigma) \equiv (\sigma^2 \vee \varphi)/\varphi^2$. Consequently, studying $n_{a,T}$ reduces to analyzing

$$\varphi_{a,t} \equiv \varphi(n_{a,t}; \overline{\sigma}_a) = \frac{\overline{\sigma}_a \vee n_{a,t}^{-1/2}}{\sqrt{n_{a,t}}}, \qquad a \in [K], \ t \in [T].$$

Here the two pieces in $\varphi$ match the variance-driven term ($\sigma/\sqrt{n}$) and the boundedness-driven term ($1/n$) in *Bernstein's inequality*; the rescalings $\overline{\sigma}_a$ and $\overline{\Delta}_a$ place variance and gap on the same Bernstein-type exploration scale used by our UCB-V bonus.

---

[1] [4] does not directly provide the worst-case regret bound for UCB-V, and the best known worst-case regret bound for UCB-V is claimed as $\mathcal{O}(\sqrt{KT \log T})$ in [28]. In Appendix E.2, we show that the $\mathcal{O}(\sqrt{\sum_{a \neq a^\star} \sigma_a^2 T \log T})$ regret for UCB-V can be derived based on results in [4].

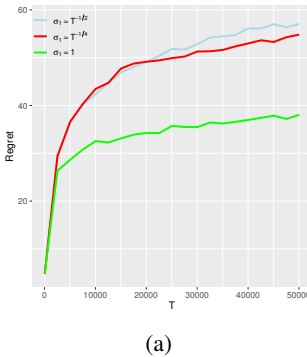
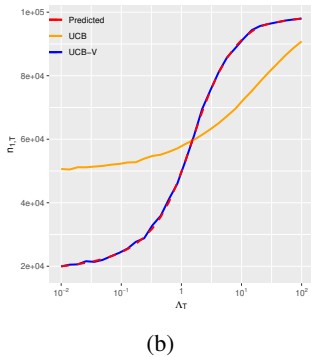

(a)                                                                  (b)

Figure 2: (a): The regrets of UCB-V with $\sigma_2 \asymp T^{-1/4}, \Delta_2 \asymp 1/\sqrt{T}$ fixed and $\sigma_1 \asymp T^{-1/2}, T^{-1/4}, 1$, each instance with $10$ repetitions. (b): The median and $30\%$ quantile of $n_{1,T}$ (optimal arm-pulling count) for UCB and UCB-V, under varying $\Lambda_T$ in the $\sigma_1 = \mathfrak{o}(\sigma_2)$ regime, with $T = 1,000,000$ over $30$ repetitions for each $\Delta_2$. The red dotted line is the predicted $n_{1,T}$ of UCB-V using (6). A more detailed numerical setting is provided in Appendix H.

## 2.1  Implicit bounds for arm-pulling numbers

Consider Algorithm 1 with $K = 2$. We start with the following concentration result for $\varphi_{a,T}$.

**Proposition 1.** *Recall that $\overline{\sigma}_a \equiv \overline{\sigma}_a(\rho; T)$ and $\overline{\Delta}_a \equiv \overline{\Delta}_a(\rho; T)$ for $a \in [2]$. Fix any $\delta > 0, \rho > 1$, and any positive integer $T \geq 4$ such that $\varepsilon \equiv \varepsilon(\delta; \rho; T) \equiv 5\big(\sqrt{48 \frac{\log(\log(T/\delta)/\delta)}{\rho \log T}} + \frac{128 \log(1/\delta)}{\rho \log T}\big)^{1/2} + \frac{200}{\sqrt{T}} \leq \frac{1}{2}$. Then, with probability at least $1 - \delta$, we have*

$$\left(\frac{1-\varepsilon}{1+\varepsilon}\right)^4 \leq \frac{\overline{\sigma}_1^2 \vee \varphi_{1,T}}{T\varphi_{1,T}^2} + \frac{\overline{\sigma}_2^2 \vee (\varphi_{1,T} + \overline{\Delta}_2)}{T(\varphi_{1,T} + \overline{\Delta}_2)^2} \leq \left(\frac{1+\varepsilon}{1-\varepsilon}\right)^4$$

*and $\left(\frac{1-\varepsilon}{1+\varepsilon}\right)^4 \leq \frac{\varphi_{2,T}}{\varphi_{1,T}+\overline{\Delta}_2} \leq \left(\frac{1+\varepsilon}{1-\varepsilon}\right)^4$.*

The proof of Proposition 1 above is inspired by the delicate analysis of bonus terms for the canonical UCB [23] in the large-$T$ regime and is presented in Appendix C.1. Proposition 1 establishes a sandwich inequality, demonstrating that as term $\varepsilon(\delta; \rho; T)$ approaches $0$, the concerned ratio converges to $1$. This can be viewed as a non-asymptotic and variance-aware extension of the results found in [21] and [23] for the canonical UCB. To incorporate variance information, we have also developed a Bernstein-type non-asymptotic law of iterated logarithm result in Lemma 8 in Appendix G, which is of independent interest. We now make several comments regarding Proposition 1:

**Deriving the asymptotic equation.** When selecting $\delta = (\log T)^{-1}$, we find that for any $\rho > 0$, the $\varepsilon(\delta; \rho; T)$ term approaches $0$ at a rate of $\mathcal{O}\big(\frac{\log \log T}{\log T}\big)$. Under this selection, Proposition 1 yields the following probability convergence guarantee

$$\frac{\overline{\sigma}_1^2 \vee \varphi_{1,T}}{T\varphi_{1,T}^2} + \frac{\overline{\sigma}_2^2 \vee (\varphi_{1,T} + \overline{\Delta}_2)}{T(\varphi_{1,T} + \overline{\Delta}_2)^2} = 1 + \mathfrak{o}_{\mathbf{P}}(1). \tag{1}$$

It is noteworthy that in this asymptotic result, the $\mathfrak{o}_{\mathbf{P}}(1)$ term converges to $0$ at a relatively slow rate, both in probability and magnitude. For each $T$, it has a probability of at least $1 - \mathcal{O}\big(\frac{1}{\log T}\big)$ of being bounded by $\mathcal{O}\big(\frac{\log \log T}{\log T}\big)$. This slow rate appears not only in our result but also in [21] and [23], necessitating a considerably large $T$ to observe theoretical predictions clearly in experiments, as illustrated in Figure 2b.

**Permissible values of $\rho$ for high probability bounds.** One popular scale selection of $\delta$ is $\delta \asymp 1/T^2$, where Proposition 1 becomes a high-probability guarantee widely adopted in the pure-exploration and regret analysis literature [5, 3]. With such selection, the requirement $\varepsilon(\delta; \rho; T) < 1/2$ translates to $\rho > c'$ for some universal and sufficiently large constant $c'$. Thus, we have that with probability at

least $1 - \mathcal{O}(1/T^2)$,

$$1 - (c')^{-1} \leq \frac{\overline{\sigma}_1^2 \vee \varphi_{1,T}}{T\varphi_{1,T}^2} + \frac{\overline{\sigma}_2^2 \vee (\varphi_{1,T} + \overline{\Delta}_2)}{T(\varphi_{1,T} + \overline{\Delta}_2)^2} \leq 1 + (c')^{-1}. \tag{2}$$

In particular, since the centered term is a monotone increasing function of $\varphi_{1,T}$, this result provides a high-probability confidence region for $\varphi_{1,T}$ and consequently for $n_{1,T}$, as shown in Figure 3a in Appendix D.1. These high probability bounds of $n_{1,T}$ enable us to derive the refined regret bound for UCB-V in Section 4.

The implications outlined above will be made precise in the subsequent two sections.

## 3 Asymptotic characterization of arm-pulling numbers

### 3.1 Stability of the asymptotic equation

Consider the deterministic equation

$$f(\varphi) = 1 \quad \text{with} \quad f(\varphi) \equiv \frac{\overline{\sigma}_1^2 \vee \varphi}{T\varphi^2} + \frac{\overline{\sigma}_2^2 \vee (\varphi + \overline{\Delta}_2)}{T(\varphi + \overline{\Delta}_2)^2}. \tag{3}$$

Recall that $\overline{\sigma}_a \equiv \sigma_a/\sqrt{\rho \log T}$ and $\overline{\Delta}_a \equiv \Delta_a/(\rho \log T)$ for $a = 1, 2$, so the equation actually depends on both $\rho$ and $T$. For any fixed $T \in \mathbb{N}_+$ and $\rho > 1$, Proposition 1 indicates that $\varphi_{1,T}$ satisfies $f(\varphi_{1,T}) = 1 + \zeta$, where $\zeta$ represents a perturbation term. Given the asymptotic equation derived in (1), it is reasonable to conjecture that the behavior of $\varphi_{1,T}$ aligns with the solution of the above deterministic equation. To formally connect $\varphi_{1,T}$ to the solution of (3), we conduct a perturbation analysis of this equation. Specifically, we have the following result.

**Lemma 1.** *Fix any $T \in \mathbb{N}_+$ and $\rho > 1$. It holds that:*

1. *The fixed-point equation (3) admits a unique solution $\varphi^\star \in (1/T, 1)$.*

2. *Assume that there exist some $\zeta \in (-1/2, 1/2)$ and $\varphi_\zeta$ such that $f(\varphi_\zeta) = 1 + \zeta$. Then*

$$\left| \frac{\varphi_\zeta}{\varphi^\star} - 1 \right| \leq 27|\zeta| \cdot \left( 1 + \left| \frac{\overline{\sigma}_2^2}{T\overline{\Delta}_2^2} - 1 \right|^{-1} \wedge \frac{\overline{\sigma}_2}{\overline{\sigma}_1 \vee T^{-1/2}} \right). \tag{4}$$

The proof of Lemma 1 above is provided in Appendix D.1. Intuitively, Lemma 1 asserts that in either the homogeneous variance case ($\sigma_2 \asymp \sigma_1$) or when the ratio $\overline{\sigma}_2^2/(T\overline{\Delta}_2^2)$ is bounded away from 1, the solution $\varphi_\zeta$ of the $\zeta$-perturbed equation is provably bounded by $(1 \pm \mathcal{O}(\zeta))\varphi^\star$. However, when $\sigma_1 = \mathfrak{o}(\sigma_2)$ and the ratio $\overline{\sigma}_2^2/(T\overline{\Delta}_2^2)$ approaches 1, the stability guarantee of the lemma breaks down. Although this result may seem pessimistic—as $\overline{\sigma}_2^2/(T\overline{\Delta}_2^2) \sim 1$ causes the right-hand side of (4) to diverge—it accurately reflects the behavior of the perturbed solution. This is illustrated in Figure 3b in Appendix D.1, where instability arises when $\overline{\sigma}_2^2/(T\overline{\Delta}_2^2) = 1$.

### 3.2 Asymptotic stability

One important consequence of the perturbation bound is the following asymptotic stability result.

**Theorem 1.** *Consider Algorithm 1 with $K = 2$. Assume that*

$$\varlimsup_{T \to \infty} \left\{ \left| \frac{\overline{\sigma}_2^2}{T\overline{\Delta}_2^2} - 1 \right|^{-1} \wedge \frac{\overline{\sigma}_2}{\overline{\sigma}_1 \vee T^{-1/2}} \right\} < \infty. \tag{5}$$

*Then for $a \in [2]$, we have $n_{a,T}/n_{a,T}^\star \xrightarrow{p} 1$, where $n_{1,T}^\star$ is the unique solution to equation*

$$\frac{\overline{\sigma}_1^2 \vee \varphi(n_{1,T}^\star; \overline{\sigma}_1)}{T\varphi^2(n_{1,T}^\star; \overline{\sigma}_1)} + \frac{\overline{\sigma}_2^2 \vee (\varphi(n_{1,T}^\star; \overline{\sigma}_1) + \overline{\Delta}_2)}{T(\varphi(n_{1,T}^\star; \overline{\sigma}_1) + \overline{\Delta}_2)^2} = 1 \tag{6}$$

*and $n_{2,T}^\star \equiv \overline{\sigma}_2^2 \vee (\varphi(n_{1,T}^\star; \overline{\sigma}_1) + \overline{\Delta}_2)/(\varphi(n_{1,T}^\star; \overline{\sigma}_1) + \overline{\Delta}_2)^2$.*

The proof of Theorem 1 above is provided in Appendix D.2, relying on Proposition 1 and Lemma 1. Lemma 1 shows that the boundedness condition in (5) is essential for ensuring the stability of the arm-pulling process, as defined in Definition 1. Specifically, when $\sigma_1 = \sigma_2 = 1$, condition (5) is always satisfied, and the asymptotic equation (6) reduces to the canonical UCB setting studied in [21] and [23], where stability is guaranteed. The key insight behind such stability in an even more general homogeneous setting $\sigma_1 \asymp \sigma_2$ is that the optimal arm's pulling number grows linearly with $T$, as noted in [23, Eqn. (22)]. In the inhomogeneous case where $\sigma_1 = \mathfrak{o}(\sigma_2)$, the boundedness of $|\overline{\sigma}_2^2/(T\overline{\Delta}_2^2) - 1|^{-1}$ becomes crucial for stability and appears to be *novel*, ensuring stability even when $n_{1,T}$ grows sub-linearly in $T$. The instability result in the next subsection complements Theorem 1 by presenting a counterexample where the boundedness condition in (5) fails. An extension to the $K$-armed setting is discussed in Appendix B.

**Asymptotic behavior of arm-pulling numbers.** Let us write $\varphi^\star \equiv \varphi(n_{1,T}^\star; \overline{\sigma}_1)$ for simplicity. From Theorem 1, the asymptotic behavior of $n_{1,T}$ can be derived through $n_{1,T}^\star$, which, in turn, can be fully determined by $\varphi^\star$, provided that the boundedness condition in (5) holds. Thus, understanding the analytical properties of $\varphi^\star$ is sufficient to determine the asymptotic behavior of $n_{1,T}$.

The function $f(\varphi^\star)$ is known to be monotonic increasing in $\varphi^\star$, and the solution to $f(\varphi^\star) = 1$ lies within the interval $[0, 1]$. This allows us to efficiently compute the numerical behavior of $\varphi^\star$ to predict both asymptotic and finite-time arm-pulling behavior, as illustrated in Figure 3a in Appendix D.1. However, due to the presence of a maximum operation and the complexity of the underlying equation, obtaining a closed-form solution for $\varphi^\star$ remains difficult. Below, we present several extreme cases that highlight *new* phenomena arising from the incorporation of variance information, which differs from the classical UCB algorithm. A full analytical characterization of $\varphi^\star$ is left for future research.

**Example 1.** *When $\sigma_1, \sigma_2 = \Omega(1)$, Eqn. (6) simplifies to*

$$\frac{n_{1,T}^\star}{T} + \left( \frac{\sigma_1}{\sigma_2} \cdot \sqrt{\frac{T}{n_{1,T}^\star}} + \frac{\Delta_2}{\sigma_2} \sqrt{\frac{T}{\rho \log T}} \right)^{-2} = 1.$$

*In the moderate gap regime $\Delta_2 \sim \sigma_2 \sqrt{\theta \log T/T}$ for some fixed $\theta \in \mathbb{R}_{\geq 0}$, we have $n_{1,T}^\star/T = \lambda^\star(\theta)\big(1 + \mathfrak{o}(1)\big)$, with $\lambda^\star(\theta)$ being the unique solution to equation $\lambda + \big( \frac{\sigma_1}{\sigma_2} \cdot \frac{1}{\sqrt{\lambda}} + \sqrt{\theta/\rho} \big)^{-2} = 1$. Moreover, we can compute the limits $\lim_{\theta \to +\infty} \lambda^\star(\theta) = 1$ and $\lim_{\theta \to 0} \lambda^\star(\theta) = \frac{\sigma_1^2}{\sigma_1^2 + \sigma_2^2}$.*

*This result recovers the asymptotic equation for the canonical UCB in [21] and [23] when $\sigma_1 = \sigma_2 = 1$. The $\lambda^\star$ equation derived here can be viewed as an extension of those in [21]. Notably, in the small-gap limit $\theta \to 0$, the arm-pulling allocation for UCB-V becomes proportional to the variance instead of being equally divided.*

**Example 2.** *When $\sigma_1 \leq \sqrt{\rho \log T/T}, \sigma_2 = \Omega(1)$, Eqn. (6) simplifies to*

$$\frac{n_{1,T}^\star}{T} + \left( \frac{1}{\sigma_2} \sqrt{\frac{\rho \log T}{T}} \cdot \frac{T}{n_{1,T}^\star} + \frac{\Delta_2}{\sigma_2} \sqrt{\frac{T}{\rho \log T}} \right)^{-2} = 1.$$

*In the moderate gap regime where $\Delta \sim \sigma_2 \sqrt{\theta \log T/T}$ for some fixed $\theta \in \mathbb{R}_{\geq 0} \setminus \{\rho\}$, we have $n_{1,T}^\star/T = \lambda^\star(\theta)\big(1 + \mathfrak{o}(1)\big)$, with $\lambda^\star(\theta)$ being the unique solution to equation $\lambda + \big( \frac{1}{\sigma_2} \sqrt{\frac{\rho \log T}{T}} \cdot \frac{1}{\lambda} + \sqrt{\theta/\rho} \big)^{-2} = 1$. Moreover, we can compute the limits $\lim_{\theta \to +\infty} \lambda^\star(\theta) = 1$ and $\lim_{\theta \to 0} \lambda^\star(\theta) = \frac{2}{\sqrt{1 + 4\overline{\sigma}_2^2 T} + 1}$.*

*This limit indicates a distinct behavior between UCB-V and UCB: as $\theta \to 0$, the number of pulls for the optimal arm becomes sub-linear in $T$, specifically $\mathcal{O}(\sqrt{T}/\overline{\sigma}_2)$, while for UCB, it approaches $T/2$. Additionally, we have a more detailed description of the transition between sub-linear and linear pulling times: (i) When $\theta < \rho$, $\lambda^\star(\theta) \leq 1/(\overline{\sigma}_2(1 - \sqrt{\theta/\rho})\sqrt{T})$ and (ii) When $\theta > \rho$, $\lambda^\star(\theta) \geq 1 - \rho/\theta$. This result indicates that the transition from $\mathcal{O}(\sqrt{T})$ pulling time to $\Omega(T)$ pulling time for the optimal arm occurs at $\theta = \rho$.*

**Inference with UCB-V.** Another key implication of Theorem 1 is its relevance to the post-policy inference in the UCB-V algorithm. To begin, we recall the notion of arm stability as defined in [23, Definition 2.1].

**Definition 1.** *An arm $a \in [K]$ is stable if there exists a deterministic sequence $n_{a,T}^\star$ such that*

$$\frac{n_{a,T}}{n_{a,T}^\star} \xrightarrow{p} 1 \quad with \quad n_{a,T}^\star \to \infty, \tag{7}$$

*where $n_{a,T}^\star$ may depend on $T$, $\{\mu_a\}_{a\in[K]}$, and $\{\sigma_a^2\}_{a\in[K]}$.*

This notion of stability guarantees that, as $T \to \infty$, the number of times that an arm is pulled becomes predictable, thus enabling valid statistical inference for each arm's reward distribution. Specifically, under the stability condition, the following $Z$-statistic converges in distribution to a standard normal

$$\frac{\sqrt{n_{a,T}}}{\widehat{\sigma}_{a,T}} \left( \bar{X}_{a,T} - \mu_a \right) \Rightarrow \mathcal{N}(0, 1). \tag{8}$$

This result is crucial for constructing valid hypothesis tests and confidence intervals in the adaptive sampling settings [29, 39]. For space efficiency, we provide a more detailed discussion along with simulation results in Appendix I.

### 3.3 Unstable results: closer look and hard instances

Recall that Theorem 1 establishes the stability result, except for the case when $\sigma_1 = \mathfrak{o}(\sigma_2)$ and $\overline{\sigma}_2^2/(T\overline{\Delta}_2^2) = 1$ hold simultaneously. In this section, we provide a hard instance in this setting and show the instability result. More precisely, let us consider the setting similar to Example 2, with $\sigma_2 = 1$, $\sigma_1 = 0$, and gap $\Delta_2 = \sigma_2\sqrt{\theta \log T/T}$.

In this scenario, the behavior of the solution to the asymptotic equation is described by $n_{1,T}^\star \sim \lambda^\star(\theta)T$, where $\lambda^\star(\theta)$ solves $\lambda + \left(\sqrt{\rho \log T/T}/\lambda + \sqrt{\theta/\rho}\right)^{-2} = 1$ or equivalently,

$$\frac{1}{\sqrt{1-\lambda}} - \frac{1}{\lambda}\sqrt{\frac{\rho \log T}{T}} = \sqrt{\theta/\rho}. \tag{9}$$

To heuristically explain why $\theta = \rho$ (or equivalently $\overline{\sigma}_2^2/(T\overline{\Delta}_2^2) = 1$) acts as a phase transition point and leads to instability, we formally take the first-order expansion $1/\sqrt{1-\lambda} \approx 1 + \lambda/2$ in (9) and multiply both sides by $\lambda$. This yields a quadratic equation in $\lambda$, $\frac{1}{2}\lambda^2 + (1 - \sqrt{\theta/\rho})\lambda - \sqrt{\frac{\rho \log T}{T}} = 0$. Notably, the order of the solution in $T$ depends crucially on the sign of $1 - \sqrt{\theta/\rho}$. For any $\varepsilon > 0$, by observing the analytical solution to the quadratic equation, we have

$$\lambda = \begin{cases} \mathcal{O}(\varepsilon^{-1}\sqrt{\frac{\rho \log T}{T}}) & \text{when } 1 - \sqrt{\theta/\rho} \geq \varepsilon, \\ \Omega(\varepsilon) & \text{when } 1 - \sqrt{\theta/\rho} \leq -\varepsilon. \end{cases}$$

In particular, a perturbation in $\theta$ around $\theta = \rho$ of magnitude $\varepsilon$ with different signs can lead to a fluctuation in $n_{1,T}^\star$ from $\mathcal{O}(\varepsilon^{-1}\sqrt{T \log T})$ to $\Omega(\varepsilon T)$.

Based on the above intuition and informal argument, we can rigorously demonstrate the *instability* result by constructing a Bernoulli bandit instance and establishing a time-uniform anti-concentration result for the Bernoulli reward process via Donsker's principle. Intuitively, our anti-concentration result shows that, with constant probability, $\theta$ can be perturbed by a magnitude of $\mathcal{O}((\log T)^{-1})$ with different signs, which then implies instability in $n_{1,T}$ even when we play UCB-V with the oracle information of the variance (i.e., with $\widehat{\sigma}_{a,t} = \sigma_a$). We summarize the instability result in the following proposition and provide its proof in Appendix D.3.

**Proposition 2.** *Consider the following two-armed Bernoulli bandit instance: for $\mu = 1/2$ and $\Delta^2 = \mu(1-\mu)\rho \log T/T$, arm 1 has reward $\mu + \Delta$ and variance 0, and arm 2 has reward $\mu$ and variance $\mu(1-\mu)$. Assume that $\sigma_a$'s are known and let $n_{a,T}$ be computed by Algorithm 1 with $\widehat{\sigma}_{a,t} = \sigma_a$. Then for sufficiently large $T$, there exist some constants $c_0, c_1 > 0$ such that*

$$\mathbb{P}(n_{1,T} \leq c_1\sqrt{T}\log T) \wedge \mathbb{P}(n_{1,T} \geq c_1^{-1}T/\log^{1/2} T) > c_0.$$

**Remark 1.** *In the example above we set $\sigma_1 = 0$ for simplicity. The instability analysis in Proposition 2 also extends to cases where the optimal arm has variance $\sigma_1 = \mathcal{O}(T^{-1/2})$, since in this regime the*

*UCB-V bonus for the optimal arm matches the $\sigma_1 = 0$ case. By contrast, constructing an instance with $\sigma_1 = T^{-\alpha}$ for some $\alpha \in (0, 1/2)$ is more delicate and appears to require new ideas: the associated asymptotic equation becomes substantially more involved, and a complete analysis is currently unclear.*

The existence of an unstable instance at $\Lambda_T = 1$ *complements* our previous finding, where UCB-V's stability was shown for $\Lambda_T \neq 1$. This result highlights a *contrast* between the UCB-V and canonical UCB: as shown in [21] and [23], the canonical UCB is stable for all $\Delta$, which corresponds to the case in our setting when $\sigma_1 = \sigma_2 = 1$. Our result demonstrates that, in a *heterogeneous variance* environment, UCB-V may exhibit significant fluctuations. Another implication of this instability is that, unlike the canonical UCB, the CLT for the $Z$-statistic may *not* hold for data collected by UCB-V, as shown in Figure 4b in Appendix I. This necessitates the developments of new statistical inference methods for UCB-V collected data or new variance-aware decision-making algorithms with stronger stability guarantees than UCB-V, which we leave as a valuable future direction.

## 4 Refined regret for variance-aware decision making

In this section, we show that a refined regret can be achieved by UCB-V based on our arm-pulling number bounds presented in Section 2. Previously, the best-known regret for UCB-V, shown in [4], is given by [2]

$$\text{Reg}(T) \leq \max_{\Delta_2 \geq 0} C \log T \Big( \frac{\sigma_2^2}{\Delta_2} + 2 \Big) \wedge \Delta_2 T \leq C' \sigma_2 \sqrt{T \log T}. \tag{11}$$

As mentioned earlier, the regret in (11) does not account for the effect of $\sigma_1$, which contradicts the empirical performance of UCB-V and is conservative in the large $\sigma_1$, small $\sigma_2$ regime, as shown in Figure 2a. On the other hand, the derived asymptotic equation for the arm-pulling times naturally indicates a dependency of $n_{2,T}$ on $\sigma_1$, opening the possibility for refined worst-case regret bounds. More precisely, by adopting the high-probability bound in (2), we can show the following proposition.

**Proposition 3.** *There exists some universal constant $C_0$ such that with $\rho \geq C_0$ and $T \geq 3$,*

$$\mathbb{P} \left( n_{2,T} \leq \frac{16}{\overline{\Delta}_2 + \varphi(n_{1,T}; \overline{\sigma}_1)} \vee \Big( \frac{16\sigma_2}{\overline{\Delta}_2 + \varphi(n_{1,T}; \overline{\sigma}_1)} \Big)^2 \right) \geq 1 - \frac{6}{T^2}.$$

By applying the upper bound on the number of pulls for sub-optimal arms, we are able to derive the refined worst-case regret for UCB-V.

**Theorem 2.** *There exists some universal constant $C_0 > 0$ such that for $\rho \geq C_0$ and $T \geq 3$, we have that with probability at least $1 - 6/T^2$,*

$$\text{Reg}(T) \leq 16 \Big( \sigma_2 \wedge \frac{16\sigma_2^2}{\sigma_1} \Big) \sqrt{\rho T \log T} + \sqrt{\rho \log T}.$$

The proof of the above results can be found in Appendix E. The regret bound in Theorem 2 above improves upon the bound in (11) for the large $\sigma_1$, small $\sigma_2$ regime, while recovering (11) in the small $\sigma_1$ regime. We note that sharper bounds may be derived in special instances based on Proposition 3. For example, in the Bernoulli setting where $\mu_1$ is close to 1, the variance inequality $\sigma_2^2 \lesssim \sigma_1^2 + \Delta_2$ holds. In this case, Proposition 3 implies a regret bound of the form $\mathcal{O}(\sigma_1 \sqrt{T \log T})$, which matches the result established in [30].

**Regret results for the $K$-armed setting.** Since most works consider the general $K$-armed setting, for ease of comparison we now presenting the $K$-armed extension of Theorem 2, we leave the

---

[2]Rather than presenting an equation of the form (11), [4] established the following gap-dependent upper bound on the number of times a suboptimal arm is pulled:

$$\mathbb{E}(n_{2,T}) \leq C \log T \left( \frac{\sigma_2^2}{\Delta_2^2} + \frac{2}{\Delta_2} \right). \tag{10}$$

However, it is straightforward to show (11) by combining (10) with the trivial bound $\text{Reg}(T) \leq \Delta_2 \mathbb{E}(n_{2,T})$.

rigorous statements in Theorem 5 in Appendix B. For a $K$-armed MAB problem with variances $\{\sigma_a^2\}_{a=1}^K$(assuming again W.L.O.G. arm 1 is optimal), we have:

$$\mathrm{Reg}(T) = \mathcal{O}\left(\sqrt{\sum_{a \in [2:K]} \sigma_a^2} \wedge \left(\sum_{a \in [2:K]} \frac{\sigma_a^2}{\sigma_1}\right) \cdot \sqrt{T \log T}\right),$$

The best known worst-case regret of UCB-V was previously known as $\mathcal{O}(\sqrt{KT \log T})$, as discussed in [28]. In comparison, our result shows that UCB-V can achieve regret adaptive to both the variances of sub-optimal and optimal arms. Beyond UCB-V, for Thompson sampling algorithms, the $\mathcal{O}(\sqrt{\sum_{a \in [2:K]} \sigma_a^2 T \log T})$ regret was proved in a Bayesian setting, where each arm's reward distribution follows a known prior distribution. Especially, even when working with the Bayesian regret, the dependency on $\sigma_1$ is *previously unrevealed*. We believe that our results for UCB-V can be extended to these posterior sampling algorithms.

**Optimality of Theorem 2.** Now we establish a matching lower bound to show the optimality of Theorem 2. To describe the lower bound result, consider any given distributions $\mathbf{P}_1$ and $\mathbf{P}_2$. Let $\boldsymbol{v} = (\mathbf{P}_1, \mathbf{P}_2)$ denote a 2-armed bandit instance, where $\mathbf{P}_i$ represents the distribution of the $i$-th arm. To emphasize the dependency on the means $\mu_i \equiv \mathbb{E}_{X \sim \mathbf{P}_i}[X]$ and variances $\sigma_i^2 \equiv \mathrm{Var}_{X \sim \mathbf{P}_i}[X]$ of the arms, we redundantly express this instance as $\boldsymbol{v} = (\mathbf{P}_1(\mu_1, \sigma_1), \mathbf{P}_2(\mu_2, \sigma_2))$.

Given any instance $\boldsymbol{v}$, we use the notation $\sigma_{\mathsf{opt}}^2(\boldsymbol{v}), \sigma_{\mathsf{sub}}^2(\boldsymbol{v})$ to denote variances of optimal and sub-optimal arms under $\boldsymbol{v}$ for clarity. For any policy $\pi$ and any instance $\boldsymbol{v}$, we denote $\mathbb{E}_{\pi\boldsymbol{v}}$ and $\mathbb{P}_{\pi\boldsymbol{v}}$ as the expectation and probability with respect to the reward distribution induced by $\pi$ under $\boldsymbol{v}$. More precisely, we establish the following regret lower bounds. The proof is deferred to Appendix E.3.

**Theorem 3.** *Given any sufficiently large $T > 0$ and $0 < \sigma^2 < 1/3$, consider the following class of problems with $\sigma$-bounded variances:*

$$\mathcal{V}_\sigma \equiv \{(\mathbf{P}_1(\mu_1, \sigma_1), \mathbf{P}_2(\mu_2, \sigma_2)) : \sigma_1 \vee \sigma_2 \leq \sigma, \mathrm{supp}(\mathbf{P}_i(\mu_i, \sigma_i)) \subset [0, 1], i \in \{1, 2\}\}.$$

*Then for every policy $\pi$, there exists some $\boldsymbol{v} \in \mathcal{V}$ such that,*

$$\mathbb{E}_{\pi\boldsymbol{v}} \mathrm{Reg}(T) = \Omega(\sigma_{\mathsf{sub}}(\boldsymbol{v})\sqrt{T}). \tag{12}$$

*Moreover, the following trade-off lower bound holds: Given any $0 < \beta < 1/2$ and $c_T = \mathcal{O}(\mathrm{poly}(\log T))$, consider*

$$\mathcal{V}'_\beta \equiv \{(\mathbf{P}_1(\mu_1, \sigma_1), \mathbf{P}_2(\mu_2, \sigma_2)) \in \mathcal{V}_\sigma : \mu_1 > \mu_2, \sigma_1 = T^\beta \sigma_2^2, \sigma_1 \geq T^{-\beta}/256\}$$

*and the good policy class*

$$\Pi_{\mathsf{good}} \equiv \left\{\pi : \max_{\boldsymbol{v} \in \mathcal{V}} \frac{\mathbb{E}_{\pi\boldsymbol{v}} \mathrm{Reg}(T)}{\sigma_{\mathsf{sub}}(\boldsymbol{v})} \leq c_T \sqrt{T}\right\}.$$

*Then for any $\pi \in \Pi_{\mathsf{good}}$, there exists some $\boldsymbol{v}' \in \mathcal{V}'_\beta$ such that*

$$\mathbb{E}_{\pi\boldsymbol{v}'} \mathrm{Reg}(T) = \Omega\left(\frac{\sigma_{\mathsf{sub}}^2(\boldsymbol{v}')}{\sigma_{\mathsf{opt}}(\boldsymbol{v}')} \sqrt{\frac{T}{c_T}}\right). \tag{13}$$

Theorem 3 states that, in the general scenario with $\sigma$-bounded variances, no algorithm can achieve a regret better than $\sigma_{\mathsf{sub}}(\boldsymbol{v})\sqrt{T}$. This reveals the optimality of Theorem 2 in the regime where $\sigma_{\mathsf{opt}}(\boldsymbol{v}) \leq \sigma_{\mathsf{sub}}(\boldsymbol{v})$. Furthermore, we show that in $\sigma_{\mathsf{opt}} > \sigma_{\mathsf{sub}}$ regime, any *reasonably good* algorithm that performs nearly optimal in worst-case over $\mathcal{V}$ (i.e., the algorithms lie within $\Pi_{\mathsf{good}}$) cannot achieve a regret better than $\frac{\sigma_{\mathsf{sub}}^2}{\sigma_{\mathsf{opt}}}\sqrt{T}$. In particular, the regret upper bound in Theorem 2 demonstrates that UCB-V matches such trade-off lower bound in the regime where $\sigma_{\mathsf{opt}} > \sigma_{\mathsf{sub}}$, illustrating its optimality.

**Conclusion.** In this paper, we provide a refined analysis of the UCB-V algorithm, including a precise characterization of its asymptotic arm-pulling behavior and high-probability, non-asymptotic bounds that lead to a sharper and optimal regret upper bound. Several valuable future directions remain open. First, the instability result is established only under the regime $\sigma_1 = \mathfrak{o}(\sqrt{\log T/T})$ and $\sigma_2 = \Omega(1)$, while the stability condition in the more general case $\sigma_1 = \mathfrak{o}(\sigma_2)$ is not yet sharply understood. Second, as discussed in Section B, our stability condition in the $K$-armed setting requires a uniform-type separation condition $\min_a \bar{\Delta}_a \geq \bar{\sigma}_a \sqrt{(K-1)/T}$, which we believe can be further refined.

**Acknowledgments.** This work is generously supported by NSF CCF-2312205, ONR 13983263, and 2026 New York University Center for Global Economy and Business grant.

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

# A   Additional related works

**Variance-aware decision making**

In the multi-armed bandit setting, leveraging the variance information to balance the exploration-exploitation trade-off was first studied by [5] and [3, 4], where the UCB-V algorithm was proposed and analyzed. Beyond the optimistic approach, [28] examined elimination-based algorithms that utilize variance information, while [20] and [33] focused on analyzing the performance of Thompson sampling in variance-aware contexts.

Beyond the MAB model, variance information has been utilized in contextual bandit and Markov Decision Process settings [36, 40, 9, 41, 10, 37]. Among these works, [40] and [41] are most relevant to our study, especially regarding the linear contextual bandits. Instead of assuming that each arm $a$ has a fixed variance $\sigma_a$, they assumed that at each round $t$, all arms have the same variance $\sigma_t$ depending on the time-step $t$. In the homogeneous case – i.e. when $\sigma_a \equiv \sigma, \sigma_t \equiv \sigma$ for some $\sigma > 0$ – their settings coincides with ours, and both algorithms achieve a regret of $\sqrt{\sigma^2 KT}$ regret. However, in the general case, the regret results are not directly comparable.

**Asymptotic behavior analysis in multi-armed bandits**

Our investigation into the asymptotic behaviors of UCB-V is inspired by recent advancements in the precise characterization of arm-pulling behavior [21, 23], which focused on the canonical UCB algorithm. Beyond the canonical UCB algorithm, another line of research explores the asymptotic properties of bandit algorithms within the Bayesian frameworks, particularly under diffusion scaling [14, 2, 24], where reward gaps scale as $1/\sqrt{T}$. Additionally, a noteworthy body of work [15, 16, 34] conducts asymptotic analyses in the regime described by [25], where the reward gaps $\Delta$ remain constant as $T$ increases.

**Inference with adaptively collected data**

In addition to the regret minimization, there is a growing amount of interest in statistical inference for the bandit problems. A significant body of research addresses the online debiasing and adaptive inference methods [11, 7, 8, 12], but our findings are more closely related to studies focused on the post-policy inference [29, 39, 19, 38]. These works aim to provide valid confidence intervals for reward-related quantities based on data collected from pre-specified adaptive policies. Among them, the most relevant works are [21] and [23]. The former addresses the inference guarantee of $Z$-estimators collected by UCB in the two-armed setting, while the latter extends this to the $K$-armed setting. In particular, these two works assert the stability of the pulling time for the optimal arm under arbitrary gap conditions $\Delta$, enabling the application of the martingale Central Limit Theorem (CLT) to obtain an asymptotically normal estimator of $\mu^\star$. However, our results point out a critical regime where such stability *breaks down*, as shown in Figure 4b, highlighting the price of utilizing variance information to enhance the regret minimization performance.

# B   Discussions on $K$-armed setting

In this section, we extend the results in Theorem 1, which addresses the 2-armed setting, to the general $K$-armed case. Recall that $\overline{\sigma}_a \equiv \overline{\sigma}_a(\rho; T)$ and $\overline{\Delta}_a \equiv \overline{\Delta}_a(\rho; T)$ for $a \in [K]$. The formal statement of the result is given in the following theorem.

**Theorem 4.** *Consider Algorithm 1. Assume that for any $a \in [2 : K]$,*

$$\varlimsup_{T \to \infty} \left\{ \max_{\mathfrak{a} \in [2:K]} \left( \frac{1}{K-1} - \frac{\overline{\sigma}_{\mathfrak{a}}^2}{T\overline{\Delta}_{\mathfrak{a}}^2} \right)_+^{-1} \wedge \left( \frac{\overline{\sigma}_a^2}{T\overline{\Delta}_a^2} - 1 \right)_+^{-1} \wedge \frac{\overline{\sigma}_a}{\overline{\sigma}_1 \vee T^{-1/2}} \right\} < \infty. \tag{14}$$

*Then for $a \in [K]$, we have*

$$\frac{n_{a,T}}{n_{a,T}^\star} \xrightarrow{p} 1,$$

*where $n_{1,T}^\star$ is the unique solution to equation*

$$\sum_{a\in[K]} \frac{\overline{\sigma}_a^2 \vee (\varphi(n_{1,T}^\star;\overline{\sigma}_1)+\overline{\Delta}_a)}{T(\varphi(n_{1,T}^\star;\overline{\sigma}_1)+\overline{\Delta}_a)^2} = 1 \tag{15}$$

*and for $a \in [2:K]$,*

$$n_{a,T}^\star \equiv \frac{\overline{\sigma}_a^2 \vee (\varphi(n_{1,T}^\star;\overline{\sigma}_1)+\overline{\Delta}_a)}{(\varphi(n_{1,T}^\star;\overline{\sigma}_1)+\overline{\Delta}_a)^2}.$$

When $K = 2$, the condition in (4) simplifies to the stability condition for the 2-armed setting as stated in Theorem 1. While instability occurs when (4) is violated in the 2-armed case (cf. Proposition 2), there is no clear reason to expect that (4) is optimal for the $K$-armed setting. Indeed, as indicated in the proof, the fixed-point equation (15) may remain stable if there exists *some* $a \in [K] \setminus \{1\}$ that satisfies the boundedness condition in (14). However, formally characterizing such an $a$ poses a significant challenge in itself. As such, a full generalization to the $K$-armed setting is left for future work.

**Asymptotic arm-pulling characterization (moderate gap, $K$-armed).**   To build intuition for (15), consider the $K$-armed setting with all variances $\{\sigma_a\}_{a\in[K]} = \Omega(1)$ and gaps in the moderate-gap regime $\Delta_a = \sigma_a\sqrt{(\theta_a \log T)/T}$ with fixed $\theta_a \geq 0$. In this regime the variance branches of $\varphi(\cdot;\cdot)$ are active, and

$$\varphi(n_{1,T}^\star;\overline{\sigma}_1) = \frac{\sigma_1}{\sqrt{n_{1,T}^\star}\sqrt{\rho\log T}}, \qquad \varphi(n_{1,T}^\star;\overline{\sigma}_1)+\overline{\Delta}_a = \frac{\sigma_1/\sqrt{n_{1,T}^\star}+\sigma_a\sqrt{\theta_a/T}}{\sqrt{\rho\log T}}.$$

Plugging these into (15) and multiplying both sides by $\rho\log T$ yields

$$\sum_{a\in[K]} \frac{\sigma_a^2}{T\big(\sigma_1/\sqrt{n_{1,T}^\star}+\sigma_a\sqrt{\theta_a/T}\big)^2} = 1, \qquad \frac{n_{a,T}^\star}{T} = \frac{\sigma_a^2}{T\big(\sigma_1/\sqrt{n_{1,T}^\star}+\sigma_a\sqrt{\theta_a/T}\big)^2}.$$

As $\theta_a \to 0$ (shrinking gaps), the normalized pulling rates converge to

$$\frac{n_{a,T}^\star}{T} \longrightarrow \frac{\sigma_a^2}{\sum_{j\in[K]}\sigma_j^2}.$$

That is, when the gaps vanish, each arm's asymptotic pulling rate becomes proportional to its variance, generalizing the two-armed intuition and the homogeneous-variance case.

Besides the stability results, the high-probability bounds for $n_{a,T}$ in the $K$-armed setting, as in (2), still hold when (6) is replaced with (15), as shown in Appendix C. The following corollary provides a refined regret bound in the $K$-armed setting.

**Theorem 5.** *There exists some universal constant $C_0 > 0$ such that for $\rho \geq C_0$ and $T \geq 3$, we have that with probability at least $1 - 3K/T^2$,*

$$\mathrm{Reg}(T) \leq 16\left(\sqrt{\sum_{a\in[2:K]}\sigma_a^2} \wedge \sum_{a\in[2:K]}\frac{16\sigma_a^2}{\sigma_1}\right)\sqrt{\rho T\log T} + (K-1)\sqrt{\rho\log T}.$$

When $K = 2$, Theorem 5 above aligns with Theorem 2. A comparison with previous findings is provided at the end of the previous section. Notably, in the small $\sigma_a^2/\sigma_1$ regime, the bound in Theorem 5 grows linearly with the number of arms $K$. We conjecture that the result in Theorem 5 is optimal up to logarithmic factors for variance-aware decision-making, even for more complicated algorithms beyond UCB-V. Establishing the corresponding minimax lower bound is left for future work.

# C Proofs for Section 2

For any $T \in \mathbb{N}_+$ and $\delta > 0$, we define events

$$\mathcal{M}(\delta; \rho; T) \equiv \left\{ |\bar{X}_{a,t} - \mu_a| \leq \varphi_{a,t} \cdot \mathsf{e}(\delta; \rho; T), \ \forall t \in [T], a \in [K] \right\},$$

$$\mathcal{V}(\delta; \rho; T) \equiv \left\{ |\widehat{\sigma}_a^2 - \sigma_a^2| \leq \varphi_{a,t} \cdot \mathsf{e}(\delta; \rho; T), \ \forall t \in [T], a \in [K] \right\},$$

where $\mathsf{e}(\delta; \rho; T) \equiv \sqrt{48\rho \log T \log(\log(T/\delta)/\delta)} + 128 \log(1/\delta)$. The majority of the technical analysis in this paper will be done on event $\mathcal{M}(\delta; \rho; T) \cap \mathcal{V}(\delta; \rho; T)$. It is worth noting that, as shown in Proposition 7 below, we have $\mathbb{P}(\mathcal{M}(\delta; \rho; T) \cap \mathcal{V}(\delta; \rho; T)) \geq 1 - 3K\delta$ for any $\delta < 1/2$. Although the statements in Section 2 are presented for the 2-armed setting, the proof in this section applies to the $K$-armed setting as well.

## C.1 Proof of Proposition 1

Recall that $\varphi_{a,t} \equiv \varphi(n_{a,t}; \overline{\sigma}_a)$. For simplicity, denote by $\widehat{\varphi}_{a,t} \equiv \varphi(n_{a,t}; \widehat{\sigma}_{a,t}/\sqrt{\rho \log T})$ for $a \in [K]$ and $t \in [T]$. The following lemma provides a quantified error between $\widehat{\varphi}_{a,t}$ and $\varphi_{a,t}$.

**Lemma 2.** *On event $\mathcal{M}(\delta; \rho; T) \cap \mathcal{V}(\delta; \rho; T)$, we have that for any $a \in [K]$ and $t \in [T]$,*

$$|\widehat{\varphi}_{a,t} - \varphi_{a,t}| \leq \varphi_{a,t} \cdot \sqrt{\frac{\mathsf{e}(\delta; \rho; T)}{\rho \log T}}.$$

*Proof.* By the elementary inequality that $|\sqrt{x} - \sqrt{y}| \leq \sqrt{|x - y|}$ for any $x, y \geq 0$, we can derive on event $\mathcal{M}(\delta; \rho; T) \cap \mathcal{V}(\delta; \rho; T)$ that

$$|\varphi_{a,t} - \widehat{\varphi}_{a,t}| \leq \frac{|\sigma_a - \widehat{\sigma}_{a,t}|}{\sqrt{\rho \log T \cdot n_{a,t}}} = \varphi_{a,t} \cdot \left( \frac{|\sigma_a - \widehat{\sigma}_{a,t}|}{\sqrt{\rho \log T}(\overline{\sigma}_a \vee n_{a,t}^{-1/2})} \right)$$

$$\leq \varphi_{a,t} \cdot \frac{\sqrt{\varphi(n_{a,t}; \overline{\sigma}_a) \cdot \mathsf{e}(\delta; \rho; T)}}{\sqrt{\rho \log T}(\overline{\sigma}_a \vee n_{a,t}^{-1/2})} \leq \varphi_{a,t} \cdot \sqrt{\frac{\mathsf{e}(\delta; \rho; T)}{\rho \log T}},$$

which completes the proof. $\square$

For any $\delta, \rho > 0$, let $\mathsf{I}_{\pm}(\delta; \rho; T) \equiv 1 \pm \mathsf{Err}^{1/2}(\delta; \rho; T) \pm \mathsf{Err}(\delta; \rho; T)$ with $\mathsf{Err}(\delta; \rho; T) \equiv \mathsf{e}(\delta; \rho; T)/(\rho \log T)$. Recall that $\varphi_{a,T} = \varphi(n_{a,T}; \overline{\sigma}_a)$. The following proposition provides (tight) upper and lower bounds for $\varphi_{a,T}$.

**Proposition 4.** *Fix any $T \in \mathbb{N}_+$ and any $\delta, \rho \in \mathbb{R}_+$ such that $\mathsf{I}_{\pm}(\delta; \rho; T) > 0$. Then on event $\mathcal{M}(\delta; \rho; T) \cap \mathcal{V}(\delta; \rho; T)$, we have that for any $a \in [K]$,*

$$\frac{\mathsf{I}_+(\delta; \rho; T)}{\mathsf{I}_-(\delta; \rho; T)} \cdot \frac{n_{a,T}}{n_{a,T} - 1} \geq \frac{\overline{\Delta}_a + \varphi_{1,T}}{\varphi_{a,T}} \geq \frac{\mathsf{I}_-(\delta; \rho; T)}{\mathsf{I}_+(\delta; \rho; T)} \cdot \frac{n_{1,T} - 1}{n_{1,T}}. \tag{16}$$

*Proof.* The proof primarily follows the framework in [26, 23]. In the proof, we write $\mathsf{I}_{\pm} \equiv \mathsf{I}_{\pm}(\delta; \rho; T)$, $\mathcal{M} \equiv \mathcal{M}(\delta; \rho; T)$, and $\mathcal{V} \equiv \mathcal{V}(\delta; \rho; T)$ for simplicity. For any $a \in [K]$, denote by $T_a$ the last time-step such that arm $a$ was pulled.

(**Step 1**). In this step, we provide an upper bound for $\varphi_{a,T}^{-1}$. By the UCB rule, we have

$$\bar{X}_{a,T_a} + \widehat{\varphi}_{a,T_a} \cdot \rho \log T \geq \bar{X}_{1,T_a} + \widehat{\varphi}_{1,T_a} \cdot \rho \log T.$$

By Lemma 2, it holds on event $\mathcal{M} \cap \mathcal{V}$ that

$$\mathsf{I}_+ \cdot \varphi_{a,T_a} \geq \overline{\Delta}_a + \varphi_{1,T_a} \cdot \mathsf{I}_-.$$

Using further $n_{a,T_a} = n_{a,T} - 1$, $n_{1,T_a} \leq n_{1,T}$, and $\overline{\Delta}_a \geq 0$, we can show that

$$\mathsf{I}_+ \cdot \varphi(n_{a,T} - 1; \overline{\sigma}_a) \geq (\overline{\Delta}_a + \varphi_{1,T}) \cdot \mathsf{I}_-.$$

As $\varphi(n_{a,T} - 1; \overline{\sigma}_a) \leq \varphi_{a,T} n_{a,T}/(n_{a,T} - 1)$, we can obtain an upper bound for $\varphi_{a,T}^{-1}$

$$\frac{\mathsf{l}_+}{\mathsf{l}_-} \geq \frac{\overline{\Delta}_a + \varphi_{1,T}}{\varphi_{a,T}} \cdot \frac{n_{a,T} - 1}{n_{a,T}}. \tag{17}$$

(**Step 2**). In this step, we provide a lower bound for $\varphi_{a,T}^{-1}$. Consider the last time-step such that arm 1 was pulled. In view of the UCB rule, it holds that

$$\bar{X}_{a,T_1} + \widehat{\varphi}_{a,T_1} \cdot \rho \log T \leq \bar{X}_{1,T_1} + \widehat{\varphi}_{1,T_1} \cdot \rho \log T.$$

By Lemma 2, we have that on event $\mathcal{M} \cap \mathcal{V}$,

$$\mathsf{l}_- \cdot \varphi_{a,T_1} \leq \overline{\Delta}_a + \varphi_{1,T_1} \cdot \mathsf{l}_+. \tag{18}$$

Then by $n_{a,T_1} \leq n_{a,T} - 1, n_{1,T_1} \leq n_{1,T} - 1$, and $\overline{\Delta}_a \geq 0$, it follows that

$$\mathsf{l}_- \cdot \varphi_{a,T} \leq \left(\overline{\Delta}_a + \varphi(n_{1,T} - 1; \overline{\sigma}_1)\right) \cdot \mathsf{l}_+, \tag{19}$$

As $\varphi(n_{1,T} - 1; \overline{\sigma}_1) \leq \varphi_{1,T} n_{1,T}/(n_{1,T} - 1)$, we can obtain a lower bound for $\varphi_{a,T}^{-1}$

$$\frac{\mathsf{l}_-}{\mathsf{l}_+} \leq \frac{\overline{\Delta}_a + \varphi_{1,T}}{\varphi_{a,T}} \cdot \frac{n_{1,T}}{n_{1,T} - 1}. \tag{20}$$

Combining (17) and (20) above concludes the proof. $\qquad\square$

**Proposition 5.** *Fix any $T \in \mathbb{N}_+$ and any $\delta, \rho \in \mathbb{R}_+$ such that $\mathsf{l}_\pm(\delta; \rho; T) > 0$. Then on event $\mathcal{M}(\delta; \rho; T) \cap \mathcal{V}(\delta; \rho; T)$, we have that for any $a \in [K]$,*

$$\left(\frac{\mathsf{l}_+(\delta; \rho; T)}{\mathsf{l}_-(\delta; \rho; T)} \cdot \frac{n_{a,T}}{n_{a,T} - 1}\right)^{-1} \leq \frac{\overline{\sigma}_a \vee n_{a,T}^{-1/2}}{\overline{\sigma}_a \vee (\varphi_{1,T} + \overline{\Delta}_a)^{1/2}} \leq \left(\frac{\mathsf{l}_-(\delta; \rho; T)}{\mathsf{l}_+(\delta; \rho; T)} \cdot \frac{n_{1,T} - 1}{n_{1,T}}\right)^{-1}.$$

*Proof.* In the proof below, let us write $\mathsf{l}_\pm \equiv \mathsf{l}_\pm(\delta; \rho; T)$ for simplicity. Denote by

$$\xi_u \equiv \frac{\mathsf{l}_+}{\mathsf{l}_-} \cdot \frac{n_{a,T}}{n_{a,T} - 1}, \quad \xi_l \equiv \frac{\mathsf{l}_-}{\mathsf{l}_+} \cdot \frac{n_{1,T} - 1}{n_{1,T}}.$$

Note that for any $\sigma > 0$, function $\varphi(\cdot; \sigma) : \mathbb{R}_+ \to \mathbb{R}_+$ is bijective and monotone decreasing, with its inverse map given by

$$\psi(z; \sigma) : z \to \frac{z \vee \sigma^2}{z^2}, \quad z \in \mathbb{R}_+.$$

Starting from the sandwich inequality (16), the monotonicity of $\varphi$ yields that

$$\psi\left(\xi_u^{-1}(\overline{\Delta}_a + \varphi_{1,T}); \overline{\sigma}_a\right) \geq n_{a,T} \geq \psi\left(\xi_l^{-1}(\overline{\Delta}_a + \varphi_{1,T}); \overline{\sigma}_a\right).$$

As $\xi_l^{-1} > 1$, we have

$$n_{a,T}^{-1} \leq \frac{\xi_l^{-2}(\overline{\Delta}_a + \varphi_{1,T})^2}{\xi_l^{-1}(\overline{\Delta}_a + \varphi_{1,T}) \vee \overline{\sigma}_a^2} \leq \frac{\xi_l^{-2}(\overline{\Delta}_a + \varphi_{1,T})^2}{(\overline{\Delta}_a + \varphi_{1,T}) \vee \overline{\sigma}_a^2}.$$

Using further the fact that $\psi(z; \sigma)^{-1} \vee \sigma^2 = z \vee \sigma^2$, we arrive at

$$n_{a,T}^{-1} \vee \overline{\sigma}_a^2 \leq \left((\overline{\Delta}_a + \varphi_{1,T}) \vee \overline{\sigma}_a^2\right) \cdot \xi_l^{-2}.$$

Similar, from $\xi_u^{-1} < 1$ we can derive that

$$n_{a,T}^{-1} \vee \overline{\sigma}_a^2 \geq \left((\overline{\Delta}_a + \varphi_{1,T}) \vee \overline{\sigma}_a^2\right) \cdot \xi_u^{-2}.$$

Combining the above two bounds completes the proof. $\qquad\square$

**Lemma 3.** *Fix any $T \in \mathbb{N}_+$ and any $\delta, \rho \in \mathbb{R}_+$ such that $0 < \mathsf{l}_+(\delta; \rho; T)/\mathsf{l}_-(\delta; \rho; T) \leq 2$ and $T > 2K$. Then on event $\mathcal{M}(\delta; \rho; T) \cap \mathcal{V}(\delta; \rho; T)$, we have that for any $a \in [K]$,*

$$n_{a,T} \geq \frac{\sigma_1^2 T \vee \sqrt{T}}{100K} \cdot \mathbf{1}_{a=1} + \frac{\rho \log T}{8} \cdot \mathbf{1}_{a \neq 1}. \tag{21}$$

*Proof.* In the proof below, we write $\mathsf{I}_\pm \equiv \mathsf{I}_\pm(\delta; \rho; T)$ for simplicity. Let us first consider the lower bound for $n_{1,T}$. Suppose by contradiction that (21) does not hold and we have

$$n_{1,T}^{-1/2} > \frac{10\sqrt{K}}{(\overline{\sigma}_1 \sqrt{T}) \vee T^{1/4}},$$

which implies $\overline{\sigma}_1 \vee n_{1,T}^{-1/2} \geq \overline{\sigma}_1 \vee (\overline{\sigma}_1^{-1} T^{-1/2} \wedge T^{-1/4}) = \overline{\sigma}_1 \vee T^{-1/4}$. Thus, it follows that

$$\varphi_{1,T} = (\overline{\sigma}_1 \vee n_{1,T}^{-1/2}) n_{1,T}^{-1/2} \geq \frac{10\sqrt{K}}{\sqrt{T}}.$$

On the other hand, as $\sum_{a \in [K]} n_{a,T} = T$, there must exist some $a \in [K], a \neq 1$ such that $n_{a,T} \geq T/K$. For such $a$, we have by (17) that

$$\frac{\mathsf{I}_+}{\mathsf{I}_-} \cdot \frac{n_{a,T}}{n_{a,T} - 1} \geq \frac{\sqrt{n_{a,T}} \cdot \varphi_{1,T}}{\overline{\sigma}_1 \vee n_{a,T}^{-1/2}} \Rightarrow \frac{2T}{T - K} \geq \frac{\sqrt{n_{a,T}} \cdot \varphi_T}{\overline{\sigma}_1 \vee n_{a,T}^{-1/2}}$$

$$\Rightarrow \frac{2T}{T - K} \cdot \frac{\overline{\sigma}_1 \vee n_{a,T}^{-1/2}}{\sqrt{n_{a,T}}} \geq \frac{10\sqrt{K}}{\sqrt{T}} \Rightarrow \frac{2T\sqrt{K}}{T - K} \cdot \frac{1}{\sqrt{T}} \geq \frac{10\sqrt{K}}{\sqrt{T}},$$

which then leads to a contradiction. The lower bound for $n_{1,T}$ follows.

Next, we prove the lower bound for $n_{a,T}, a \in [2 : K]$. Starting from (20), we have by the lower bound for $n_{1,T}$ in the first part that

$$n_{a,T} \geq \frac{1}{2(\overline{\Delta}_a + \varphi_{1,T})} \cdot \frac{n_{1,T} - 1}{n_{1,T}} \geq \frac{1}{4} \left( \frac{1}{\rho \log T} + \frac{10\sqrt{K}}{T^{1/4}} \right)^{-1} \geq \frac{\rho \log T}{8},$$

which proves the claim. $\qquad\square$

We are now ready to prove Proposition 1.

*Proof Proposition 1.* In the proof below, let us write $\mathsf{I}_\pm \equiv \mathsf{I}_\pm(\delta; \rho; T)$ and $\mathsf{Err} \equiv \mathsf{Err}(\delta; \rho; T)$ for simplicity. It follows from Propositions 4 and 5 that on event $\mathcal{M}(\delta; \rho; T) \cap \mathcal{V}(\delta; \rho; T)$,

$$\frac{\mathsf{I}_-^2}{\mathsf{I}_+^2} \cdot \frac{(n_{a,T} - 1)(n_{1,T} - 1)}{n_{a,T} n_{1,T}} \leq \frac{\sqrt{n_{a,T}} \cdot (\overline{\Delta}_a + \varphi_{1,T})}{\overline{\sigma}_a \vee (\varphi_{1,T} + \overline{\Delta}_a)^{1/2}} \leq \frac{\mathsf{I}_+^2}{\mathsf{I}_-^2} \cdot \frac{n_{a,T} n_{1,T}}{(n_{a,T} - 1)(n_{1,T} - 1)}.$$

Applying the lower bounds for $n_{a,T}$ in Lemma 3, we can further bound the right-hand side above as

$$\frac{\mathsf{I}_+^2}{\mathsf{I}_-^2} \cdot \left( 1 - \frac{8}{\rho \log T} \right)^{-1} \cdot \left( 1 - \frac{100K}{\sqrt{T}} \right)^{-1}$$

$$\leq \left( \frac{1 + 2\sqrt{\mathsf{Err}}}{1 - 2\sqrt{\mathsf{Err}}} \right)^2 \left( 1 - \frac{8}{\rho \log T} \right)^{-1} \left( 1 - \frac{100K}{\sqrt{T}} \right)^{-1} \leq \frac{(1 + 2\sqrt{\mathsf{Err}})^2}{1 - 5\sqrt{\mathsf{Err}} - 100KT^{-1/2}},$$

where we have used $\sqrt{\mathsf{Err}} \geq 8/(\rho \log T)$ and the elementary inequality $(1 - x)(1 - y)(1 - z) \geq 1 - x - y - z$ for $\forall x, y, z > 0$.

Using the same argument for the left-hand side, we can show that

$$\frac{1 - 5\sqrt{\mathsf{Err}} - 100KT^{-1/2}}{(1 + 2\sqrt{\mathsf{Err}})^2} \leq \frac{\sqrt{n_{a,T}} \cdot (\overline{\Delta}_a + \varphi_{1,T})}{\overline{\sigma}_a \vee (\varphi_{1,T} + \overline{\Delta}_a)^{1/2}} \leq \frac{(1 + 2\sqrt{\mathsf{Err}})^2}{1 - 5\sqrt{\mathsf{Err}} - 100KT^{-1/2}}.$$

Summing over $a$ and using the fact that $\sum_{a \in [K]} n_{a,T} = T$, we can obtain that

$$\frac{(1 - 5\sqrt{\mathsf{Err}} - 100KT^{-1/2})^2}{(1 + 2\sqrt{\mathsf{Err}})^4} \leq \sum_{a=1}^K \frac{\overline{\sigma}_a^2 \vee (\varphi_{1,T} + \overline{\Delta}_a)}{T(\overline{\Delta}_a + \varphi_{1,T})^2} \leq \frac{(1 + 2\sqrt{\mathsf{Err}})^4}{(1 - 5\sqrt{\mathsf{Err}} - 100KT^{-1/2})^2},$$

as desired. $\qquad\square$

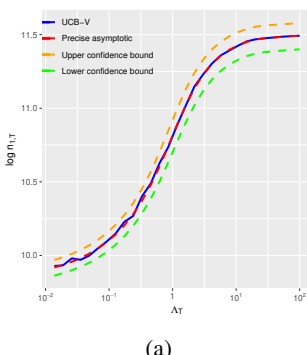
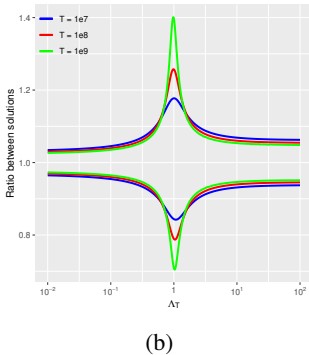

|(a)|(b)|

Figure 3: (a): The confidence region of $n_{1,T}$ under different $\Lambda_T$, with $\sigma_1 = \mathfrak{o}(\sigma_2)$. The dotted lines represent the exact and perturbed solutions of (3), where the perturbed curves solve $f(\varphi) = 1 \pm 1/\log T$. The UCB-V line shows the number of arm pulls under the UCB-V algorithm with $30\%$ quantile, with $T = 10^5$ over 30 repetitions. (b): The ratio between the perturbed solution $f(\varphi) = 1 \pm 1/\log T$ and the exact solution $f(\varphi) = 1$ is shown for $T = 10^5, 10^7, 10^9$ under different $\Lambda_T$. It can be seen that the ratio deviates from 1 as $\Lambda_T \to 1$ with increasing $T$.

# D Proofs for Section 3

## D.1 Proof of Lemma 1

*Proof of Lemma 1.* (1). First, we prove the existence and uniqueness of $\varphi^\star$. Note that $f(\varphi)$ is continuously monotone decreasing on $(0, \infty)$. On the other hand, as

$$f(1/T) = \frac{\overline{\sigma}_1^2 \vee T^{-1}}{T^{-1}} + \frac{\overline{\sigma}_2^2 \vee (T^{-1} + \overline{\Delta}_2)}{T(T^{-1} + \overline{\Delta}_2)^2} > \frac{\overline{\sigma}_1^2 \vee T^{-1}}{T^{-1}} \geq 1$$

and

$$f(1) = \frac{\overline{\sigma}_1^2 \vee 1}{T} + \frac{\overline{\sigma}_2^2 \vee (1 + \overline{\Delta}_2)}{T(1 + \overline{\Delta}_2)^2} \leq \frac{2}{T} < 1,$$

there must exist a unique $\varphi^\star \in (1/T, 1)$ such that $f(\varphi^\star) = 1$, proving the claim.

(2). We only show the proof for the case of $\zeta > 0$, while the case of $\zeta < 0$ can be handled similarly. Note by the monotonicity, we have $\varphi_\zeta \leq \varphi^\star$. Let $\varphi_\zeta/\varphi^\star = 1 - \overline{\zeta}$ for some $\overline{\zeta} \geq 0$ depending on $\zeta$. Our aim is to derive an upper bound for $\overline{\zeta}$.

It follows from (3) that at least one of the following holds:

$$\textbf{Case 1}: \frac{\overline{\sigma}_1^2 \vee \varphi^\star}{T(\varphi^\star)^2} \geq \frac{1}{2}, \quad \textbf{Case 2}: \frac{\overline{\sigma}_2^2 \vee (\varphi^\star + \overline{\Delta}_2)}{T(\varphi^\star + \overline{\Delta}_2)^2} \geq \frac{1}{2}.$$

**Case 1**: If $\overline{\sigma}_1^2 = \overline{\sigma}_1^2 \vee \varphi^\star$, we have $\overline{\sigma}_1^2 = \overline{\sigma}_1^2 \vee \varphi_\zeta$. Hence, it holds that

$$\zeta = f(\varphi_\zeta) - f(\varphi^\star) \geq \frac{\overline{\sigma}_1^2 \vee \varphi_\zeta}{T\varphi_\zeta^2} - \frac{\overline{\sigma}_1^2 \vee \varphi^\star}{T(\varphi^\star)^2} = \frac{\overline{\sigma}_1^2}{T(\varphi^\star)^2}\left(\frac{1 - (1 - \overline{\zeta})^2}{(1 - \overline{\zeta})^2}\right) \geq \frac{1}{2}\overline{\zeta}.$$

For the case when $\varphi^\star = \overline{\sigma}_1^2 \vee \varphi^\star$, we can compute that

$$\zeta = f(\varphi_\zeta) - f(\varphi^\star) \geq \frac{\overline{\sigma}_1^2 \vee \varphi_\zeta}{T\varphi_\zeta^2} - \frac{\overline{\sigma}_1^2 \vee \varphi^\star}{T(\varphi^\star)^2} \geq \frac{1}{T\varphi^\star}\left(1 - \frac{1}{1 - \overline{\zeta}}\right) \geq \frac{1}{2}\overline{\zeta}.$$

By combining the above results, in Case 1, we can conclude that $\overline{\zeta} \leq 2\zeta$.

**Case 2**: Note that in this case, we have

$$\frac{1}{T\varphi^\star} \vee \frac{\overline{\sigma}_1^2}{T(\varphi^\star)^2} \leq \frac{\overline{\sigma}_1^2 \vee \varphi^\star}{T(\varphi^\star)^2} \leq \frac{1}{2}. \tag{22}$$

On the other hand, if $\varphi^\star + \overline{\Delta}_2 = \overline{\sigma}_2^2 \vee (\varphi^\star + \overline{\Delta}_2)$, we can show that $1/(T\varphi^\star) \geq 1/T(\varphi^\star + \overline{\Delta}_2) \geq 1/2$. Thus, it must hold that $\varphi^\star = T/2$ and $\overline{\Delta}_2 = 0$. This implies that when $\varphi^\star + \overline{\Delta}_2 = \overline{\sigma}_2^2 \vee (\varphi^\star + \overline{\Delta}_2)$,

$$\zeta = f(\varphi_\zeta) - f(\varphi^\star) \geq \frac{1}{T(\varphi_\zeta + \overline{\Delta}_2)} - \frac{1}{T(\varphi^\star + \overline{\Delta}_2)}$$

$$= \frac{1}{T\varphi^\star}\left(1 - \frac{1}{1 - \overline{\zeta}}\right) \geq \frac{1}{2}\overline{\zeta}.$$

For the case when $\overline{\sigma}_2^2 = \overline{\sigma}_2^2 \vee (\varphi^\star + \overline{\Delta}_2)$, using (22) and the fact that $\varphi^\star \geq T^{-1}$, we can deduce that

$$\frac{\overline{\sigma}_2^2}{T(\varphi^\star + \overline{\Delta}_2)^2} \geq \frac{1}{2} \geq \frac{\overline{\sigma}_1^2 \vee \varphi^\star}{T(\varphi^\star)^2} \geq \frac{\overline{\sigma}_1^2 \vee T^{-1}}{T(\varphi^\star)^2},$$

which implies that $\varphi^\star/(\varphi^\star + \overline{\Delta}_2) \geq (\overline{\sigma}_1 \vee T^{-1/2})/\overline{\sigma}_2$. Hence, it follows that

$$\zeta = f(\varphi_\zeta) - f(\varphi^\star) \geq \frac{\overline{\sigma}_2^2}{T(\varphi_\zeta + \overline{\Delta}_2)^2} - \frac{\overline{\sigma}_2^2}{T(\varphi^\star + \overline{\Delta}_2)^2} \tag{23}$$

$$= \frac{\overline{\sigma}_2^2(\varphi^\star - \varphi_\zeta)}{T(\varphi^\star + \overline{\Delta}_2)^2} \cdot \left(\frac{\varphi_\zeta + \varphi^\star + 2\overline{\Delta}_2}{(\varphi_\zeta + \overline{\Delta}_2)^2}\right) \geq \frac{\varphi^\star \overline{\zeta}}{2(\varphi^\star + \overline{\Delta}_2)} \geq \frac{\overline{\sigma}_1 \vee T^{-1/2}}{2\overline{\sigma}_2}\overline{\zeta}.$$

This gives the bound corresponding to $\overline{\sigma}_2/(\overline{\sigma}_1 \vee T^{-1/2})$ in (4).

Next, we will derive the bound that corresponds to $|\overline{\sigma}_2^2/(T\overline{\Delta}_2^2) - 1|^{-1}$ in (4). The analysis will be further separated into two cases: (i) $1 - \overline{\sigma}_2^2/(T\overline{\Delta}_2^2) \geq 0$ and (ii) $1 - \overline{\sigma}_2^2/(T\overline{\Delta}_2^2) < 0$.

For case (i), we can show that

$$\zeta = f(\varphi_\zeta) - f(\varphi^\star) \geq \frac{\overline{\zeta}}{1 - \overline{\zeta}} \cdot \frac{\overline{\sigma}_1^2 \vee \varphi^\star}{T(\varphi^\star)^2} + \overline{\zeta} \cdot \frac{\overline{\sigma}_2^2 \varphi^\star}{T(\varphi^\star + \overline{\Delta}_2)^3}$$

$$\geq \left(\frac{\overline{\sigma}_1^2 \vee \varphi^\star}{T(\varphi^\star)^2} + \frac{\overline{\sigma}_2^2}{T(\varphi^\star + \overline{\Delta}_2)^2} - \frac{\overline{\sigma}_2^2 \overline{\Delta}_2}{T(\varphi^\star + \overline{\Delta}_2)^3}\right) \cdot \overline{\zeta} \geq \left(1 - \frac{\overline{\sigma}_2^2}{T\overline{\Delta}_2^2}\right) \cdot \overline{\zeta}.$$

For case (ii), let us recall from (23) that

$$\zeta \geq \frac{\varphi^\star \overline{\zeta}}{2(\varphi^\star + \overline{\Delta}_2)} \geq \frac{\overline{\zeta}}{6} \cdot \mathbf{1}_{\overline{\Delta}_2/\varphi^\star \leq 2} + \frac{\varphi^\star \overline{\zeta}}{2(\varphi^\star + \overline{\Delta}_2)} \cdot \mathbf{1}_{\overline{\Delta}_2/\varphi^\star > 2}. \tag{24}$$

Note that when $\overline{\Delta}_2/\varphi^\star > 2$ (or equivalently $\varphi^\star/\overline{\Delta}_2 < 1/2$),

$$\frac{\overline{\sigma}_2^2}{T(\varphi^\star + \overline{\Delta}_2)^2} \geq \frac{\overline{\sigma}_2^2}{T\overline{\Delta}_2^2}\left(1 - 3\frac{\varphi^\star}{\overline{\Delta}_2}\right) \Rightarrow \left(\frac{\overline{\sigma}_2^2}{T\overline{\Delta}_2^2} - 1\right)\frac{1}{\varphi^\star} - \frac{3\overline{\sigma}_2^2}{T\overline{\Delta}_2^3} \leq 0.$$

Using the condition that $1 - \overline{\sigma}_2^2/(T\overline{\Delta}_2^2) < 0$, we can derive that

$$(\varphi^\star)^{-1}\overline{\Delta}_2 \leq \frac{3\overline{\sigma}_2^2}{T\overline{\Delta}_2^2} \cdot \left(\frac{\overline{\sigma}_2^2}{T\overline{\Delta}_2^2} - 1\right)^{-1}.$$

As $\overline{\sigma}_2^2/(\varphi^\star + \overline{\Delta}_2)^2 \leq T$ and $\varphi^\star/\overline{\Delta}_2 < 1/2$, we can obtain that $\overline{\sigma}_2^2/(T\overline{\Delta}_2^2) \leq 9/4$, which then entails that $(\varphi^\star)^{-1}\overline{\Delta}_2 \leq 27/(2\overline{\sigma}_2^2/(T\overline{\Delta}_2^2) - 2)$. Consequently, it holds that

$$\frac{\varphi^\star \overline{\zeta}}{2(\varphi^\star + \overline{\Delta}_2)} \cdot \mathbf{1}_{\overline{\Delta}_2/\varphi^\star > 2} \geq \frac{\overline{\zeta}}{2 + 27\left(\frac{\overline{\sigma}_2^2}{T\overline{\Delta}_2^2} - 1\right)^{-1}} \cdot \mathbf{1}_{\overline{\Delta}_2/\varphi^\star > 2}.$$

Plugging this back to (24) yields that

$$\zeta \geq \frac{\overline{\zeta}}{6 + 27\left(\frac{\overline{\sigma}_2^2}{T\overline{\Delta}_2^2} - 1\right)^{-1}}.$$

By combining the above results, in Case 2, we can conclude that

$$\overline{\zeta} \leq 27\zeta \cdot \left(1 + \left|\frac{\overline{\sigma}_2^2}{T\overline{\Delta}_2^2} - 1\right|^{-1} \wedge \frac{\overline{\sigma}_2}{\overline{\sigma}_1 \vee T^{-1/2}}\right).$$

The claim then follows by combining the estimates in Cases 1 and 2 above. $\qquad\square$

## D.2 Proof of Theorem 1

**Proposition 6.** *For any positive sequences $\{\sigma_T\}_{T \geq 1}, \{n_T\}_{T \geq 1}$, and $\{\bar{n}_T\}_{T \geq 1}$, the following are equivalent:*

1. $\lim_{T \to \infty} \varphi(n_T; \sigma_T)/\varphi(\bar{n}_T; \sigma_T) = 1$.

2. $\lim_{T \to \infty} n_T/\bar{n}_T = 1$.

*Proof.* $(1) \Rightarrow (2)$. First, noticing that by the elementary inequality,

$$1 \wedge \frac{a}{b} \leq \frac{a \vee c}{b \vee c} \leq 1 \vee \frac{a}{b}, \quad \forall a, b, c > 0, \tag{25}$$

we can deduce from (1) that $\lim_{T \to \infty}(\varphi(n_T; \sigma_T) \vee \sigma_T^2)/(\varphi(\bar{n}_T; \sigma_T) \vee \sigma_T^2) = 1$. Using further the identity $n = (\sigma^2 \vee \varphi(n; \sigma))/\varphi^2(n; \sigma)$ for any $n, \sigma > 0$, we have

$$\lim_{T \to \infty} \frac{n_T}{\bar{n}_T} = \left( \lim_{T \to \infty} \frac{\varphi^2(n_T; \sigma_T)}{\varphi^2(\bar{n}_T; \sigma_T)} \right) \cdot \left( \lim_{T \to \infty} \frac{\varphi(n_T; \sigma_T) \vee \sigma_T^2}{\varphi(\bar{n}_T; \sigma_T) \vee \sigma_T^2} \right) = 1.$$

$(2) \Rightarrow (1)$. In light of (25), we see that (2) implies $\lim_{T \to \infty}(\sigma_T \vee n_T^{-1/2})/(\sigma_T \vee \bar{n}_T^{-1/2}) = 1$. Consequently, it holds that

$$\lim_{T \to \infty} \frac{\varphi(n_T; \sigma_T)}{\varphi(\bar{n}_T; \sigma_T)} = \left( \lim_{T \to \infty} \sqrt{\frac{\bar{n}_T}{n_T}} \right) \cdot \left( \lim_{T \to \infty} \frac{\sigma_T \vee n_T^{-1/2}}{\sigma_T \vee \bar{n}_T^{-1/2}} \right) = 1.$$

The claim follows. $\qquad \square$

We are now ready to prove Theorem 1.

*Proof of Theorem 1.* Let us select $\delta = 1/\log T$. In the proof, we will write $\mathsf{I}_\pm \equiv \mathsf{I}_\pm(\delta; \rho; T)$, $\mathcal{M} \equiv \mathcal{M}(\delta; \rho; T)$, and $\mathcal{V} \equiv \mathcal{V}_T(\delta; \rho; T)$ for simplicity. We also write $\varphi^\star \equiv \varphi(n_{1,T}^\star; \bar{\sigma}_1)$. From Proposition 7, we have $\mathbb{P}(\mathcal{M} \cap \mathcal{V}) \geq 1 - 3K/\log T$.

The uniqueness of $n_{1,\star}$ follows directly from Lemma 1(1) and the fact that $\varphi(\cdot; \bar{\sigma}_1)$ is bijective. It follows from Propositions 4 and 5 that on event $\mathcal{M} \cap \mathcal{V}$,

$$\frac{\mathsf{I}_-^2}{\mathsf{I}_+^2} \cdot \frac{(n_{a,T} - 1)(n_{1,T} - 1)}{n_{a,T} n_{1,T}} \leq \frac{\sqrt{n_{a,T}} \cdot (\overline{\Delta}_a + \varphi_{1,T})}{\overline{\sigma}_a \vee (\varphi_{1,T} + \overline{\Delta}_a)^{1/2}} \leq \frac{\mathsf{I}_+^2}{\mathsf{I}_-^2} \cdot \frac{n_{a,T} n_{1,T}}{(n_{a,T} - 1)(n_{1,T} - 1)}.$$

Applying Lemma 3 and using the facts that $\varphi_{1,T} = \mathfrak{o}(1)$ on $\mathcal{M} \cap \mathcal{V}$ and $\overline{\Delta}_a = \mathfrak{o}(1)$, we can show that for any $a \in [K]$, $\lim_{T \to \infty} n_{a,T} = +\infty$ on $\mathcal{M} \cap \mathcal{V}$. Hence, the inequality in the above display implies that for any $a \in [K]$,

$$n_{a,T} \cdot \frac{(\varphi_{1,T} + \overline{\Delta}_a)^2}{\overline{\sigma}_a^2 \vee (\varphi_{1,T} + \overline{\Delta}_a)} = 1 + \mathfrak{o}_{\mathbf{P}}(1). \tag{26}$$

Using further $\sum_{a \in [K]} n_{a,T}/T = 1$, we can arrive at

$$\sum_{a \in [K]} \frac{\overline{\sigma}_a^2 \vee (\varphi_{1,T} + \overline{\Delta}_a)}{T(\varphi_{1,T} + \overline{\Delta}_a)^2} = 1 + \mathfrak{o}_{\mathbf{P}}(1). \tag{27}$$

Let us consider the case when $K = 2$. By resorting to Lemma 1(2), we have that $\varphi_{1,T}/\varphi^\star = 1 + \mathfrak{o}_{\mathbf{P}}(1)$, which together with Proposition 6 implies that $n_{1,T}/n_{1,T}^\star = 1 + \mathfrak{o}_{\mathbf{P}}(1)$.

For arm 2, with (26) at hand, by Proposition 6 it suffices to show that

$$\frac{\varphi_{1,T}}{\varphi^\star} = 1 + \mathfrak{o}_{\mathbf{P}}(1) \quad \Rightarrow \quad \frac{\overline{\sigma}_2^2 \vee (\varphi^\star + \overline{\Delta}_2)}{\overline{\sigma}_2^2 \vee (\varphi_{1,T} + \overline{\Delta}_2)} \cdot \frac{(\varphi_{1,T} + \overline{\Delta}_2)^2}{(\varphi^\star + \overline{\Delta}_2)^2} = 1 + \mathfrak{o}_{\mathbf{P}}(1),$$

which is straightforward by (25). $\qquad \square$

### D.3 Proof of Proposition 2

We need the following lemma to prove Proposition 2.

**Lemma 4.** *Consider the Two-armed Bernoulli Bandit instance in Proposition 2. Recall that $T_i$ is the last time that arm $i$ was pulled. Then for sufficiently large $T$, there exists some constant $c_0 > 0$ such that*

$$\mathbb{P}\left(\mathcal{U} \equiv \left\{\bar{X}_{2,T_1} > \frac{1}{2} + \frac{1}{\sqrt{n_{2,T_1}}}\right\}\right) \wedge \mathbb{P}\left(\mathcal{L} \equiv \left\{\bar{X}_{2,T_2} < \frac{1}{2} - \frac{1}{\sqrt{n_{2,T_2}}}\right\}\right) \geq c_0.$$

*Proof.* (**Step 1**). We first prove the following: On event $\mathcal{M}_T \cap \mathcal{V}_T$, there exists some constant $0 < c < 1$ such that

$$n_{2,T_1 \wedge T_2} \geq cT. \tag{28}$$

To this end, recalling from (18) that

$$\frac{\mathsf{I}_-}{2\sqrt{n_{2,T_1}}} \leq \frac{1}{2\sqrt{T}} + \frac{\mathsf{I}_+}{n_{1,T_1}}$$

holds on event $\mathcal{M}(1/\log T; \rho; T) \cap \mathcal{V}(1/\log T; \rho; T)$ and using Lemma 3 with the fact that $n_{1,T_1} = n_{1,T} - 1$, we can deduce that for sufficiently large $T$,

$$n_{2,T_1} \gtrsim \left(\frac{1}{\sqrt{T}} + \frac{1}{n_{1,T_1}}\right)^{-2} \gtrsim T.$$

Given that $n_{2,T_2} = n_{2,T} - 1$, a similar estimate applies to $n_{2,T_2}$ if we start from (19), which proves (28).

(**Step 2**). Let $c$ be the constant in (28). It follows from Step 1 and Lemma 8 that for any $\varepsilon > 0$, there exists some $T_0 = T_0(\varepsilon) > 0$ such that for $T \geq T_0$, $\mathbb{P}(n_{2,T_1} \geq cT) \geq 1 - \varepsilon$. Using Lemma 9 and choosing $\varepsilon = c_{\mathrm{anti}}/2$, we can obtain that

$$\mathbb{P}(\mathcal{U}) \geq \mathbb{P}\left(\left\{\bar{X}_{2,T_1} > \frac{1}{2} + \frac{1}{\sqrt{n_{2,T_1}}}\right\} \cap \{n_{2,T_1} \geq \lfloor cT \rfloor\}\right)$$

$$\geq \mathbb{P}\left(\left\{\sum_{i \in [n]} X_i > \frac{n}{2} + \sqrt{n} \text{ for all } n \geq \lfloor cT \rfloor\right\} \cap \{n_{2,T_1} \geq \lfloor cT \rfloor\}\right) \geq c_{\mathrm{anti}} - \varepsilon \geq \frac{c_{\mathrm{anti}}}{2}.$$

This concludes the proof for the probability estimate for $\mathcal{U}$. The estimate for $\mathbb{P}(\mathcal{L})$ follows a similar argument, so will be omitted here to avoid repetitive details. $\qquad\square$

We are now ready to prove Proposition 2.

*Proof of Proposition 2.* (1). We will first prove the upper bound for $n_{1,T}$ on event $\mathcal{U} \cap \mathcal{M}(1/\log T; \rho; T) \cap \mathcal{V}(1/\log T; \rho; T)$. Consider time $T_1$ that arm 1 was last pulled. Using (i) the definition of $\mathcal{U}$ and (ii) the UCB rule, we can show that on event $\mathcal{U} \cap \mathcal{M}(1/\log T; \rho; T) \cap \mathcal{V}(1/\log T; \rho; T)$,

$$\mu + \frac{1}{\sqrt{n_{2,T_1}}} + \varphi_{2,T_1} \cdot \rho \log T \overset{(i)}{\leq} \bar{X}_{2,T_1} + \varphi_{2,T_1} \cdot \rho \log T \overset{(ii)}{\leq} \mu + \Delta + \frac{\rho \log T}{n_{1,T_1}}.$$

Rearranging the terms, it holds on event $\mathcal{U} \cap \mathcal{M}(1/\log T; \rho; T) \cap \mathcal{V}(1/\log T; \rho; T)$ that

$$\varphi_{2,T_1} \cdot \rho \log T \leq \Delta - \frac{1}{\sqrt{n_{2,T_1}}} + \frac{\rho \log T}{n_{1,T_1}}.$$

By the facts that $\sigma_2^2 = \mu(1-\mu)$, $\Delta^2 = \mu(1-\mu)\rho \log T/T$, $n_{2,T} \geq n_{2,T_1}$, $n_{1,T_1} = n_{1,T} - 1$, and $n_{1,T} + n_{2,T} = T$, we can arrive at

$$\frac{1}{n_{1,T} - 1} - \left(1 + \frac{1}{\rho \bar{\sigma}_2 \log T}\right) \cdot \frac{\bar{\sigma}_2}{\sqrt{T - n_{1,T}}} \geq -\frac{\bar{\sigma}_2}{\sqrt{T}}.$$

Further using $1/\sqrt{T-n_{1,T}} \geq (1+n_{1,T}/(2T))/\sqrt{T}$, the above inequality becomes

$$\frac{1}{n_{1,T}-1} - \left(1 + \frac{1}{\rho\overline{\sigma}_2\log T}\right) \cdot \frac{\overline{\sigma}_2 n_{1,T}}{2T^{3/2}} \geq \frac{1}{\rho\overline{\sigma}_2\log T} \cdot \frac{\overline{\sigma}_2}{\sqrt{T}}.$$

Note that the above inequality is quadratic, we can then solve it to obtain that

$$n_{1,T} \leq \frac{\sqrt{(\overline{B}-\overline{A})^2 + 4\overline{A}(\overline{B}+1)} - (\overline{B}-\overline{A})}{2\overline{A}},$$

where for sufficiently large $T$,

$$\overline{A} = \left(1 + \frac{1}{\rho\overline{\sigma}_2\log T}\right) \cdot \frac{\overline{\sigma}_2}{2T^{3/2}} \asymp \frac{1}{T\sqrt{T}\log T},$$

$$\overline{B} = \frac{1}{\rho\overline{\sigma}_2\log T} \cdot \frac{\overline{\sigma}_2}{\sqrt{T}} \asymp \frac{1}{\sqrt{T}\log T}.$$

Thus, we can conclude on event $\mathcal{U} \cap \mathcal{M}(1/\log T; \rho; T) \cap \mathcal{V}(1/\log T; \rho; T)$ that

$$n_{1,T} \leq \frac{2(\overline{B}+1)}{\sqrt{(\overline{B}-\overline{A})^2 + 4\overline{A}(\overline{B}+1)} + (\overline{B}-\overline{A})} \leq \frac{2(\overline{B}+1)}{\overline{B}-\overline{A}} \lesssim \sqrt{T}\log T.$$

(2). The lower bound for $n_{1,T}$ on event $\mathcal{L} \cap \mathcal{M}(1/\log T; \rho; T) \cap \mathcal{V}(1/\log T; \rho; T)$ can be established using a similar strategy. Let us consider time $T_2$ that arm 2 was last pulled. Using (i) the definition of $\mathcal{L}$ and (ii) the UCB rule, we have that on event $\mathcal{L} \cap \mathcal{M}(1/\log T; \rho; T) \cap \mathcal{V}(1/\log T; \rho; T)$,

$$\mu - \frac{1}{\sqrt{n_{2,T_2}}} + \varphi_{2,T_2} \cdot \rho\log T \overset{(i)}{\geq} \bar{X}_{2,T_2} + \varphi_{2,T_2} \cdot \rho\log T \overset{(ii)}{\geq} \mu + \Delta + \frac{\rho\log T}{n_{1,T_2}}.$$

Rearranging the terms yields that on event $\mathcal{L} \cap \mathcal{M}(1/\log T; \rho; T) \cap \mathcal{V}(1/\log T; \rho; T)$,

$$\varphi_{2,T_2} \cdot \rho\log T \geq \Delta + \frac{1}{\sqrt{n_{2,T_2}}} + \frac{\rho\log T}{n_{1,T_2}}.$$

It follows from the facts that $\overline{\Delta}_2^2 = \overline{\sigma}_2^2/T$, $n_{1,T} \geq n_{1,T_2}$, $n_{2,T_2} = n_{2,T} - 1$, and $n_{1,T} + n_{2,T} = T$ that

$$\frac{1}{n_{1,T}} - \left(1 - \frac{1}{\rho\overline{\sigma}_2\log T}\right) \cdot \frac{\overline{\sigma}_2}{\sqrt{T-n_{1,T}-1}} \leq -\frac{\overline{\sigma}_2}{\sqrt{T}}.$$

Observe that on event $\mathcal{L} \cap \mathcal{M}(1/\log T; \rho; T) \cap \mathcal{V}(1/\log T; \rho; T)$, it holds that $c_1\sqrt{T} \leq n_{1,T}+1 \leq c_2 T$ for some constants $c_1 > 0$ and $0 < c_2 < 1$, cf. Lemme 3 and (28). Then we can derive that for sufficiently large $T$,

$$\frac{1}{n_{1,T}} - \left(1 - \frac{1}{\rho\overline{\sigma}_2\log T}\right) \cdot \frac{\overline{\sigma}_2}{\sqrt{T}} \cdot \left(1 + \frac{2n_{1,T}}{(1-c_2)T}\right) \leq -\frac{\overline{\sigma}_2}{\sqrt{T}}, \tag{29}$$

where we have also used $1/\sqrt{1-(n_{1,T}+1)/T} \leq 1 + 2n_{1,T}/(1-c_2)T$. Therefore, with

$$\underline{A} = \frac{2\overline{\sigma}_2}{(1-c_2)T^{3/2}}\left(1 - \frac{1}{\rho\overline{\sigma}_2\log T}\right) \asymp \frac{1}{T\sqrt{T}\log T}, \tag{30}$$

$$\underline{B} = \frac{1}{\rho\overline{\sigma}_2\log T} \cdot \frac{\overline{\sigma}_2}{\sqrt{T}} \asymp \frac{1}{\sqrt{T}\log T}, \tag{31}$$

we can solve the inequality (29) to obtain that

$$n_{1,T} \geq \frac{\underline{B} + \sqrt{\underline{B}^2 + 4\underline{A}}}{2\underline{A}} \geq \frac{\underline{B}}{2\underline{A}} \gtrsim \frac{T}{\sqrt{\log T}}.$$

The claim now follows from Lemma 4. $\qquad\square$

# E   Proofs for Section 4

We would like to emphasize that, although the statements in Section 4 are presented for the 2-armed setting, the proof in this section applies to the $K$-armed setting as well.

## E.1   Proof of Proposition 3

We will work on the following events

$$\widetilde{\mathcal{M}}(\rho; T) \equiv \left\{ |\bar{X}_{a,t} - \mu_a| \leq \varphi_{a,t} \cdot \frac{\rho}{4} \log T, \quad \forall t \in [T], \forall a \in [K] \right\},$$

$$\widetilde{\mathcal{V}}(\rho; T) \equiv \left\{ |\widehat{\sigma}_a^2 - \sigma_a^2| \leq \varphi_{a,t} \cdot \frac{\rho}{4} \log T, \quad \forall t \in [T], \forall a \in [K] \right\}.$$

The following lemma provides a probability estimate for event $\widetilde{M}_T \cap \widetilde{V}_T$ when $\rho$ is large.

**Lemma 5.** *There exists some universal constant $C_0 > 0$ such that for $\rho \geq C_0$ and $T \geq 3$,*

$$\mathbb{P}\left( \widetilde{M}(\rho; T) \cap \widetilde{V}(\rho; T) \right) \geq 1 - 3K/T^2.$$

*Proof.* By selecting $\delta = 1/T^2$ in equations (52) and (53), we achieve the desired result, provided that $\rho \geq C_0$ with $C_0$ sufficiently large to satisfy $\sqrt{96 C_0} + 256 \leq C_0/4$. □

*Proof of Proposition 3.* First, in view of the proof for Lemma 2, it holds that on event $\widetilde{\mathcal{M}}_T \cap \widetilde{\mathcal{V}}_T$,

$$|\widehat{\varphi}_{a,t} - \varphi_{a,t}| \leq \varphi_{a,t}/2, \quad \forall a \in [K], \ \forall 1 \leq t \leq T.$$

At time-step $T_a$, by the UCB rule of $\bar{X}_{a,T_a} + \widehat{\varphi}_{a,T_a} \cdot \rho \log T \geq \bar{X}_{1,T_a} + \widehat{\varphi}_{1,T_a} \cdot \rho \log T$, we have that on event $\widetilde{\mathcal{M}}_T \cap \widetilde{\mathcal{V}}_T$,

$$\varphi_{a,T} \cdot \frac{\rho \log T}{4} + \frac{3}{2}\varphi_{a,T_a} \cdot \rho \log T \geq \Delta_a + \frac{1}{2}\varphi_{1,T} \cdot \rho \log T - \frac{\rho \log T}{4}\varphi_{1,T} \cdot \rho \log T.$$

Dividing $\rho \log T$ on both sides and rearranging the inequality, we can deduce that

$$8 \geq \frac{\overline{\Delta}_a + \varphi_{1,T}}{\varphi_{a,T_a}} \geq \frac{\overline{\Delta}_a + \varphi_{1,T}}{\varphi_{a,T}} \cdot \frac{n_{a,T} - 1}{n_{a,T}},$$

which then implies either $n_{a,T} \leq 1$ or $\varphi_{a,T} \geq (\overline{\Delta}_a + \varphi_{1,T})/16$. Therefore, we have

$$n_{a,T} \leq \frac{16}{\overline{\Delta}_a + \varphi_{1,T}} \vee \left( \frac{16\sigma_2}{\overline{\Delta}_a + \varphi_{1,T}} \right)^2.$$

The claim now follows from Lemma 5. □

## E.2   Proof of Theorem 2

*Proof of Theorem 2.* Observe that

$$\text{Reg}(T) = \sum_{a \in \mathcal{A}_1} n_{a,T} \Delta_a + \sum_{a \in \mathcal{A}_2} n_{a,T} \Delta_a$$

with $\mathcal{A}_1 \equiv \{a \in [2:K] : 16\overline{\sigma}_a^2 > \overline{\Delta}_a + \varphi_{1,T}\}$ and $\mathcal{A}_2 \equiv [2:K] \setminus \mathcal{A}_1$. It follows from Proposition 3 that

$$\mathbb{P}\left( \mathcal{E}_T \equiv \left\{ n_{2,T} \leq \frac{16}{\overline{\Delta}_2 + \varphi(n_{1,T}; \overline{\sigma}_1)} \vee \left( \frac{16\sigma_2}{\overline{\Delta}_2 + \varphi(n_{1,T}; \overline{\sigma}_1)} \right)^2 \right\} \right) \geq 1 - \frac{4}{T^2}.$$

Then for the summation on $\mathcal{A}_2$, we can derive on event $\mathcal{E}_T$ that

$$\sum_{a \in \mathcal{A}_2} n_{a,T} \Delta_a \leq \sum_{a \in \mathcal{A}_2} \frac{\Delta_a}{\varphi_{1,T} + \overline{\Delta}_a} \leq (K-1)\sqrt{\rho \log T}. \tag{32}$$

For the summation on $\mathcal{A}_1$, we have that, on one hand,

$$\sum_{a\in\mathcal{A}_1} n_{a,T}\Delta_a \leq \sqrt{\sum_{a\in\mathcal{A}_1} n_{a,T}}\sqrt{\sum_{a\in\mathcal{A}_1} n_{a,T}\Delta_a^2}$$

$$\leq 16\sqrt{T}\cdot\sqrt{\sum_{a\in\mathcal{A}_1}\frac{\overline{\sigma}_a^2}{(\overline{\Delta}_a+\varphi_{1,T})^2}\Delta_a^2} \leq 16\sqrt{\sum_{a\in\mathcal{A}_1}\sigma_a^2\cdot\rho T\log T}, \tag{33}$$

and on the other hand,

$$\sum_{a\in\mathcal{A}_1} n_{a,T}\Delta_a \leq \sum_{a\in\mathcal{A}_1}\frac{256\overline{\sigma}_a^2\Delta_a}{(\overline{\Delta}_a+\varphi_{1,T})^2} \leq \sum_{a\in\mathcal{A}_1}\frac{256\overline{\sigma}_a^2\Delta_a}{\varphi_{1,T}^2}\wedge\frac{\overline{\sigma}_a^2\Delta_a}{\overline{\Delta}_a^2}$$

$$\leq 256\sum_{a\in\mathcal{A}_1}\frac{\sigma_a^2\Delta_a T}{\sigma_1^2}\wedge\left(\frac{\sigma_a^2}{\Delta_a}\cdot\rho\log T\right) \leq 256\sum_{a\in\mathcal{A}_1}\frac{\sigma_a^2}{\sigma_1}\sqrt{\rho T\log T}. \tag{34}$$

Combining (32)–(34) above concludes the proof. $\qquad\square$

**Remark 2.** *Using the high-probability arm-pulling bound from [4]*

$$n_{a,T}\lesssim\frac{\sigma_a^2}{\Delta_a^2}+\frac{1}{\Delta_a},$$

*inequalities* (32) *and* (33) *remain valid when the definitions of $\mathcal{A}_1$ and $\mathcal{A}_2$ are replaced with*

$$\mathcal{A}_1'\equiv\{a\in[2:K]:\sigma_a^2\gtrsim\Delta_a\}\ \text{and}\ \mathcal{A}_2'\equiv[2:K]\setminus\mathcal{A}_1',$$

*respectively. This adjustment leads to a regret bound of $\mathcal{O}(\sqrt{\sum_{a\neq 1}\sigma_a^2 T\log T})$.*

### E.3 Lower bound proofs

In this section, we prove Theorem 3. For clarity, we first present the proof for the Gaussian case in Section E.3.1, then extend the construction of reward distributions to the bounded support setting in Section E.3.2.

#### E.3.1 Proof with Gaussian bandit instances

For simplicity, we present the proof with Gaussian bandit instances first and then generalize it to the hard instances constructed over bounded reward distributions.

**Theorem 6.** *Consider the following two classes of problems with any $\beta > 0$:*

$$\mathcal{V}\equiv\{(\mathcal{N}(\mu_1,\sigma_1),\mathcal{N}(\mu_2,\sigma_2)):\mu_1,\mu_2\in\mathbb{R},\sigma_1,\sigma_2\in\mathbb{R}_{\geq 0}\},$$
$$\mathcal{V}_\beta'\equiv\{(\mathcal{N}_1(\mu_1,\sigma_1),\mathcal{N}_2(\mu_2,\sigma_2))\in\mathcal{V}:\mu_1>\mu_2,\sigma_1=T^\beta\sigma_2^2,\sigma_1\geq T^{-\beta}/256\}.$$

*Then for every policy $\pi$, there exists some $\boldsymbol{v}\in\mathcal{V}$ such that*

$$\mathbb{E}_{\pi\boldsymbol{v}}\operatorname{Reg}(T)=\Omega(\sigma_{\mathsf{sub}}(\boldsymbol{v})\sqrt{T}). \tag{35}$$

*Moreover, the following trade-off lower bound holds: Given any $c_T=\mathcal{O}(\mathsf{poly}(\log T))$, consider the good policy class*

$$\Pi_{\mathsf{good}}\equiv\left\{\pi:\max_{\boldsymbol{v}\in\mathcal{V}}\frac{\mathbb{E}_{\pi\boldsymbol{v}}\operatorname{Reg}(T)}{\sigma_{\mathsf{sub}}(\boldsymbol{v})}\leq c_T\sqrt{T}\right\},$$

*then for any $\pi\in\Pi_{\mathsf{good}}$, there exists some $\boldsymbol{v}'$ such that*

$$\mathbb{E}_{\pi\boldsymbol{v}'}\operatorname{Reg}(T)=\Omega\left(\frac{\sigma_{\mathsf{sub}}^2(\boldsymbol{v}')}{\sigma_{\mathsf{opt}}(\boldsymbol{v}')}\sqrt{\frac{T}{c_T}}\right). \tag{36}$$

*Proof.* We consider the proof of (35) first. Let $\Delta \in [0, 1/2]$ and $\sigma_2 \geq \sigma_1$, where the specific values will be determined later. We consider the following two instances:

$$\boldsymbol{\nu}_1 \equiv (\mathcal{N}(\Delta, \sigma_1^2), \mathcal{N}(0, \sigma_2^2)), \quad \boldsymbol{\nu}_2 \equiv (\mathcal{N}(\Delta, \sigma_1^2), \mathcal{N}(2\Delta, \sigma_2^2)).$$

Fix a policy $\pi$. Recall the divergence decomposition Lemma for the reward distributions $\mathbb{P}_{\pi\boldsymbol{\nu}_1}, \mathbb{P}_{\pi\boldsymbol{\nu}_2}$ (see e.g. Lemma 15.1 of [27]), we have by our construction,

$$\begin{aligned}
\mathsf{KL}(\mathbb{P}_{\pi\boldsymbol{\nu}_1}\|\mathbb{P}_{\pi\boldsymbol{\nu}_2}) &= \mathbb{E}_{\pi\boldsymbol{\nu}_1}(n_{1,T})\mathsf{KL}(\mathcal{N}(\Delta, \sigma_1^2)\|\mathcal{N}(\Delta, \sigma_1^2)) \\
&\quad + \mathbb{E}_{\pi\boldsymbol{\nu}_1}(n_{2,T})\mathsf{KL}(\mathcal{N}(0, \sigma_2^2)\|\mathcal{N}(2\Delta, \sigma_2^2)) \\
&= \mathbb{E}_{\pi\boldsymbol{\nu}_1}(n_{2,T})\mathsf{KL}(\mathcal{N}(0, \sigma_2^2)\|\mathcal{N}(2\Delta, \sigma_2^2)) \leq 2T\Delta^2/\sigma_2^2.
\end{aligned} \tag{37}$$

Here, $\mathbb{P}_{\nu_i;j}$ denotes the distribution of $j$th arm in instance $\boldsymbol{\nu}_i$. Moreover, for instance $\boldsymbol{\nu}_1$, we have $\mathbb{E}_{\pi\boldsymbol{\nu}_1}\mathrm{Reg}(T) = \Delta\,\mathbb{E}_{\pi\boldsymbol{\nu}_1}n_{2,T} \geq \Delta T\mathbb{P}_{\pi\boldsymbol{\nu}_1}(n_{2,T} \geq T/2)/2$ and for instance $\boldsymbol{\nu}_2$, we have $\mathbb{E}_{\pi\boldsymbol{\nu}_2}\mathrm{Reg}(T) = \Delta\,\mathbb{E}_{\pi\boldsymbol{\nu}_2}n_{1,T} \geq \Delta T\mathbb{P}_{\pi\boldsymbol{\nu}_2}(n_{2,T} < T/2)/2$. Combining the above estimates and applying Bretagnolle-Huber inequality, we can obtain that

$$\mathbb{E}_{\pi\boldsymbol{\nu}_1}\mathrm{Reg}(T) + \mathbb{E}_{\pi\boldsymbol{\nu}_2}\mathrm{Reg}(T) \geq \frac{\Delta T}{4}\exp(-2\Delta^2 T^2/\sigma_2^2). \tag{38}$$

Taking $\Delta = \sigma_2/\sqrt{2T}$, we arrive at the lower bound

$$\mathbb{E}_{\pi\boldsymbol{\nu}_1}\mathrm{Reg}(T) \vee \mathbb{E}_{\pi\boldsymbol{\nu}_2}\mathrm{Reg}(T) \geq \frac{\sigma_2\sqrt{T}}{4\sqrt{2e}}.$$

This implies that either

$$\mathbb{E}_{\pi\boldsymbol{\nu}_1}\mathrm{Reg}(T) \geq \frac{\sigma_2\sqrt{T}}{8\sqrt{2e}} = \frac{\sigma_{\mathsf{sub}}(\boldsymbol{\nu}_1)\sqrt{T}}{8\sqrt{2e}} \quad \text{or} \quad \mathbb{E}_{\pi\boldsymbol{\nu}_2}\mathrm{Reg}(T) \geq \frac{\sigma_2\sqrt{T}}{8\sqrt{2e}} \geq \frac{\sigma_1\sqrt{T}}{8\sqrt{2e}} = \frac{\sigma_{\mathsf{sub}}(\boldsymbol{\nu}_2)\sqrt{T}}{8\sqrt{2e}}$$

hold, which completes the proof of (35).

Next, we prove the trade-off lower bound in (36). With slight abuse of notations, we construct the following instances: with $\sigma_1 \in [T^{-\beta}/256, 1]$ and $\Delta \equiv 16\sigma_1 c_T/\sqrt{T}$,

$$\boldsymbol{\nu}_1 \equiv (\mathcal{N}(\Delta, \sigma_1^2), \mathcal{N}(0, \sigma_1 T^{-\beta})), \quad \boldsymbol{\nu}_2 \equiv (\mathcal{N}(\Delta, \sigma_1^2), \mathcal{N}(2\Delta, \sigma_1 T^{-\beta})).$$

We would like to argue that for any $\pi \in \Pi_{\mathsf{good}}$,

$$\mathbb{E}_{\pi\boldsymbol{\nu}_1}\mathrm{Reg}(T) \geq \frac{\sigma_{\mathsf{sub}}^2(\boldsymbol{\nu}_1)}{256\sigma_{\mathsf{opt}}(\boldsymbol{\nu}_1)}\frac{T^{1/2}}{c_T} = \frac{T^{1/2-\beta}}{256c_T}. \tag{39}$$

We will prove this claim by contradiction. Suppose (39) does not hold, we may derive that

$$\mathbb{E}_{\pi\boldsymbol{\nu}_1}(n_{2,T}) = \mathbb{E}_{\pi\boldsymbol{\nu}_1}\mathrm{Reg}(T)/\Delta \leq \frac{T^{1-\beta}}{4096\sigma_1 c_T^2}.$$

Therefore, similar to (37), we can obtain a refined upper bound for $\mathsf{KL}(\mathbb{P}_{\pi\boldsymbol{\nu}_1}\|\mathbb{P}_{\pi\boldsymbol{\nu}_2})$ (instead of using the trivial upper bound $n_{2,T} \leq T$),

$$\mathsf{KL}(\mathbb{P}_{\pi\boldsymbol{\nu}_1}\|\mathbb{P}_{\pi\boldsymbol{\nu}_2}) \leq \frac{T^{1-\beta}}{4096\sigma_1 c_T^2}\frac{1024\sigma_1 c_T^2}{T^{1-\beta}} = \frac{1}{4}.$$

Then we may proceed as (38) to derive

$$\mathbb{E}_{\pi\boldsymbol{\nu}_1}\mathrm{Reg}(T) + \mathbb{E}_{\pi\boldsymbol{\nu}_2}\mathrm{Reg}(T) \geq 4\sigma_1 e^{-1/4}c_T\sqrt{T} \geq 3\sigma_1 c_T\sqrt{T}.$$

By the upper bound of $\mathbb{E}_{\pi\boldsymbol{\nu}_1}\mathrm{Reg}(T)$ (hypothesis), we have

$$\begin{aligned}
\mathbb{E}_{\pi\boldsymbol{\nu}_2}\mathrm{Reg}(T) &\geq 3\sigma_1 c_T\sqrt{T} - \mathbb{E}_{\pi\boldsymbol{\nu}_1}\mathrm{Reg}(T) \\
&\geq 3\sigma_1 c_T\sqrt{T} - \frac{T^{1/2-\beta}}{256c_T} \geq 3\sigma_1 c_T\sqrt{T} - \frac{\sigma_1\sqrt{T}}{c_T} \geq 2c_T\sigma_1\sqrt{T} \geq 2c_T\sigma_{\mathsf{sub}}(\boldsymbol{\nu}_2)\sqrt{T},
\end{aligned}$$

which implies that $\pi \notin \Pi_{\mathsf{good}}$, a contradiction. This proves (39) and therefore completes the proof of the trade-off lower bound in (36). □

### E.3.2 Remark on Lower Bound over Bounded Distributions

In the proof of Theorem 6, we used the Gaussian distribution for notational convenience, owing to its analytical KL divergence bound. However, the upper bound result is derived for bounded bandit instances, which leads to a slight mismatch between the lower bound and upper bound environments. In this section, we show that the proof can be readily generalized to the hard instances constructed over bounded reward distributions.

Here we specify a two-point supported variable based construction: Given any positive $\mu, \sigma^2$, we consider distributions $Q_{\mu,\sigma}$ supported over $\{0, \frac{\mu^2+\sigma^2}{\mu}\}$ with

$$Q_{\mu,\sigma}(0) = \frac{\sigma^2}{\mu^2+\sigma^2}, \quad Q_{\mu,\sigma}\Big(\frac{\mu^2+\sigma^2}{\mu}\Big) = \frac{\mu^2}{\mu^2+\sigma^2}.$$

It can be seen that $\mathbb{E}_{X\sim Q_{\mu,\sigma}}[X] = \mu$ and $\mathrm{Var}_{X\sim Q_{\mu,\sigma}}(X) = \sigma^2$.

Suppose, in addition, that $\mu < \sigma$ and let $Q_{\mu,\sigma}^{\Delta}$ denote a $\Delta$-modification of $Q_{\mu,\sigma}$ for any $0 < \Delta < \frac{\mu^2+\sigma^2}{\mu} - 2\mu$, defined as

$$Q_{\mu,\sigma}^{\Delta}(0) = \frac{\widetilde{\sigma}^2}{(\mu+\Delta)^2+\widetilde{\sigma}^2}, \quad Q_{\mu,\sigma}^{\Delta}\Big(\frac{\mu^2+\sigma^2}{\mu}\Big) = \frac{(\mu+\Delta)^2}{(\mu+\Delta)^2+\widetilde{\sigma}^2},$$

where $\widetilde{\sigma} \geq 0$ the solution to

$$\frac{(\mu+\Delta)^2+\widetilde{\sigma}^2}{\mu+\Delta} = \frac{\mu^2+\sigma^2}{\mu}. \tag{40}$$

Since $m(a) \equiv \frac{(\mu+\Delta)^2+a}{\mu+\Delta}$ is an increasing function in $a$ and

$$m(\sigma) < \frac{\mu^2+\sigma^2}{\mu} < \lim_{a\to+\infty} m(a),$$

it follows that $\widetilde{\sigma}$ is well-defined and satisfies $\widetilde{\sigma} > \sigma$.

Then we may compute the KL-divergence between $Q_{\mu,\sigma}$ and $Q_{\mu,\sigma}^{\Delta}$ as follows:

$$
\begin{aligned}
\mathsf{KL}(Q_{\mu,\sigma}\|Q_{\mu,\sigma}^{\Delta}) &= \frac{\mu^2}{\mu^2+\sigma^2}\log\frac{\frac{\mu^2}{\mu^2+\sigma^2}}{\frac{(\mu+\Delta)^2}{(\mu+\Delta)^2+\widetilde{\sigma}^2}} + \frac{\sigma^2}{\mu^2+\sigma^2}\log\frac{\frac{\sigma^2}{\mu^2+\sigma^2}}{\frac{\widetilde{\sigma}^2}{(\mu+\Delta)^2+\widetilde{\sigma}^2}} \\
&\overset{(40)}{=} \frac{\mu^2}{\mu^2+\sigma^2}\log\frac{\mu}{\mu+\Delta} + \frac{\sigma^2}{\mu^2+\sigma^2}\log\frac{\sigma^2}{\widetilde{\sigma}^2}\cdot\frac{\mu+\Delta}{\mu} \\
&\leq \frac{\mu^2}{\mu^2+\sigma^2}\log\frac{\mu}{\mu+\Delta} + \frac{\sigma^2}{\mu^2+\sigma^2}\log\frac{\mu+\Delta}{\mu} + \log\frac{\sigma^2}{\widetilde{\sigma}^2} \\
&= \frac{\sigma^2-\mu^2}{\mu^2+\sigma^2}\log\frac{\mu+\Delta}{\mu} + \log\frac{\sigma^2}{\widetilde{\sigma}^2} \\
&\leq \frac{\sigma^2-\mu^2}{\mu^2+\sigma^2}\log\frac{\mu+\Delta}{\mu},
\end{aligned}
$$

where in the last step, we have used the fact that $\log(\sigma^2/\widetilde{\sigma}^2) \leq 0$.

Taking $\mu \geq \sigma - \Delta$ in the last inequality and using the elementary inequality $\log(1+x) \leq x$ then leads to

$$\mathsf{KL}(Q_{\mu,\sigma}\|Q_{\mu,\sigma}^{\Delta}) \leq \frac{\sigma+\mu}{\mu^2+\sigma^2}\cdot\frac{\Delta^2}{\mu} \leq \frac{2\Delta^2}{\sigma(\sigma-\Delta)}.$$

Now we can summarize the above construction and calculation into the following lemma:

**Lemma 6.** *Given any $\mu, \sigma > 0$, there exists a distribution $Q_{\mu,\sigma}$ supported on $\{0, \frac{\mu^2+\sigma^2}{\mu}\}$ with*

$$\mathbb{E}_{X\sim Q_{\mu,\sigma}}[X] = \mu, \quad \mathrm{Var}_{X\sim Q_{\mu,\sigma}}[X] = \sigma^2.$$

*Given any $\Delta > 0$ and suppose that*

$$\sigma > \mu \geq \sigma - \Delta, \quad 0 < \Delta < \frac{\mu^2 + \sigma^2}{\mu} - 2\mu, \quad \Delta \leq \sigma/2 \tag{41}$$

*holds simultaneously. Then exists a $\Delta$-modification $Q_{\mu,\sigma}^\Delta$ of $Q_{\mu,\sigma}$ supported on $\{0, \frac{\mu^2 + \sigma^2}{\mu}\}$ such that*

$$\mathbb{E}_{X \sim Q_{\mu,\sigma}^\Delta}[X] = \mu + \Delta, \quad \text{Var}_{X \sim Q_{\mu,\sigma}^\Delta}[X] = (1 + \Delta/\mu)(\sigma^2 - \mu\Delta), \quad \text{KL}(Q_{\mu,\sigma} \| Q_{\mu,\sigma}^\Delta) \leq \frac{4\Delta^2}{\sigma^2}.$$

In particular, for any $\sigma > 0$, with the selection $0 < \Delta \leq \sigma/2, \mu = \sigma - \Delta$, one can verify that $(\mu, \sigma, \Delta)$ satisfies the condition (41) and the corresponding $Q_{\mu,\sigma}, Q_{\mu,\sigma}^\Delta$ are supported inside $[0, 3\sigma]$ with variances lies in $[\sigma^2, 3\sigma^2]$.

Based on Lemma 6, we now explain how to modify the distribution class constructed for proving (35) and (36) separately.

*Proof of* (12). The proof follows the same as that of (35); we highlight the differences below. Given any $\sigma_1, \sigma_2, \Delta$ satisfying $\sigma_2 \geq \sigma_1 \geq 2\Delta$, we consider the following two instances:

$$\boldsymbol{\nu}_1 \equiv \left(Q_{\sigma_2 - \Delta/2, \sigma_1}, Q_{\sigma_2 - \Delta, \sigma_2}\right), \quad \boldsymbol{\nu}_2 \equiv \left(Q_{\sigma_2 - \Delta/2, \sigma_1}, Q_{\sigma_2 - \Delta, \sigma_2}^\Delta\right).$$

Note that with such selection, Lemma 6 ensures that the distributions of the second arm's reward in both instances are supported over $[0, 3\sigma]$ with their variances lied in $[\sigma_2^2, 3\sigma_2^2]$. For the first arm, we have its variance is given by $\sigma_1^2$ and its support is bounded by

$$\frac{(\sigma_2 - \Delta/2)^2 + \sigma_1^2}{\sigma_2 - \Delta/2} \leq \frac{8\sigma_2^2}{3\sigma_2} \leq 3\sigma_2,$$

thus selecting $\sigma_2 < 1/3$ ensures that the constructed reward distributions are supported over $[0, 1]$.

Next, we can proceed the same analysis as in section E.3.1 with the proof of (35). The only difference is that the bound in (37) under our new construction turns to

$$\text{KL}(\mathbb{P}_{\pi\boldsymbol{\nu}_1} \| \mathbb{P}_{\pi\boldsymbol{\nu}_2}) = \mathbb{E}_{\pi\boldsymbol{\nu}_1}(n_{2,T}) \text{KL}(Q_{\sigma_2 - \Delta, \sigma_2} \| Q_{\sigma_2 - \Delta, \sigma_2}^\Delta) \leq 4T\Delta^2/\sigma_2^2,$$

and therefore, (38) becomes to

$$\mathbb{E}_{\pi\boldsymbol{\nu}_1} \text{Reg}(T) + \mathbb{E}_{\pi\boldsymbol{\nu}_2} \text{Reg}(T) \geq \frac{\Delta T}{4} \exp(-4\Delta^2 T^2/\sigma_2^2).$$

Taking $\Delta = \sigma_2/\sqrt{2T}$ (as $T \geq 2$, we must have that $\Delta \leq \sigma_2/2$), we arrive at the lower bound

$$\mathbb{E}_{\pi\boldsymbol{\nu}_1} \text{Reg}(T) \vee \mathbb{E}_{\pi\boldsymbol{\nu}_2} \text{Reg}(T) \geq \frac{\sigma_2\sqrt{T}}{4\sqrt{2}e^2}.$$

This implies that either

$$\mathbb{E}_{\pi\boldsymbol{\nu}_1} \text{Reg}(T) \geq \frac{\sigma_2\sqrt{T}}{8\sqrt{2}e} = \frac{\sigma_{\text{sub}}(\boldsymbol{\nu}_1)\sqrt{T}}{8\sqrt{2}e} \quad \text{or} \quad \mathbb{E}_{\pi\boldsymbol{\nu}_2} \text{Reg}(T) \geq \frac{\sigma_2\sqrt{T}}{8\sqrt{2}e} \geq \frac{\sigma_1\sqrt{T}}{8\sqrt{2}e} = \frac{\sigma_{\text{sub}}(\boldsymbol{\nu}_2)\sqrt{T}}{8\sqrt{2}e}$$

hold, which completes the proof of (12). $\qquad\square$

*Proof of* (13). The proof follows the same as that of (36); we highlight the differences below. For any fixed $0 < \beta < 1/2$ and $\sigma_1 \geq T^{-\beta}/256$, $\Delta = 16\sigma_1 c_T/\sqrt{T}$, and $\sigma_2 = \sqrt{\sigma_1 T^{-\beta}}$, we set

$$\boldsymbol{\nu}_1 \equiv \left(Q_{\sigma_2 - \Delta/2, \sigma_1}, Q_{\sigma_2 - \Delta, \sigma_2}\right), \quad \boldsymbol{\nu}_2 \equiv \left(Q_{\sigma_2 - \Delta/2, \sigma_1}, Q_{\sigma_2 - \Delta, \sigma_2}^\Delta\right).$$

In particular, notice that

$$\sigma_2/2 \geq \Delta \iff T^{-\beta}\sigma_1 \geq 512\sigma_1^2 c_T/\sqrt{T} \iff \sigma_1 \leq \frac{T^{1/2 - \beta}}{512 c_T},$$

which always holds under our selection for sufficiently large $T$. Thus the KL divergence upper bound between $Q_{\sigma_2-\Delta,\sigma}$ and $Q_{\sigma_2-\Delta,\sigma_2}^{\Delta}$ in Lemma 6 still holds under our construction. Now applying the same argument in Section E.3.1, we can arrive at

$$\mathbb{E}_{\pi\boldsymbol{\nu}_1}\mathrm{Reg}(T) + \mathbb{E}_{\pi\boldsymbol{\nu}_2}\mathrm{Reg}(T) \geq 4\sigma_1 e^{-\mathsf{KL}(\mathbb{P}_{\pi\boldsymbol{\nu}_1}\|\mathbb{P}_{\pi\boldsymbol{\nu}_2})}c_T\sqrt{T}, \quad \forall \pi \in \Pi_{\mathsf{good}}$$

when suppose in contradiction that

$$\mathbb{E}_{\pi\boldsymbol{\nu}_1}\mathrm{Reg}(T) \leq \frac{\sigma_{\mathsf{sub}}^2(\boldsymbol{\nu}_1)}{256\sigma_{\mathsf{opt}}(\boldsymbol{\nu}_1)}\frac{T^{1/2}}{c_T} = \frac{T^{1/2-\beta}}{256c_T}, \quad \forall \pi \in \Pi_{\mathsf{good}}. \tag{42}$$

Now by Lemma 6, we have

$$\mathsf{KL}(\mathbb{P}_{\pi\boldsymbol{\nu}_1}\|\mathbb{P}_{\pi\boldsymbol{\nu}_2}) \leq \frac{T^{1-\beta}}{4096\sigma_1 c_T^2}\frac{512\sigma_1^2 c_T^2}{T\sigma_2^2} = \frac{1}{8},$$

Which then, together with (42) implies

$$\mathbb{E}_{\pi\boldsymbol{\nu}_2}\mathrm{Reg}(T) \geq 4\sigma_1 e^{-1/8}c_T\sqrt{T} - \mathbb{E}_{\pi\boldsymbol{\nu}_1}\mathrm{Reg}(T) \geq 3\sigma_1 c_T\sqrt{T} - \frac{\sigma_1\sqrt{T}}{c_T}$$

$$\geq 2c_T\sigma_1\sqrt{T} \geq 2c_T\sigma_{\mathsf{sub}}(\boldsymbol{\nu}_2)\sqrt{T},$$

which implies that $\pi \notin \Pi_{\mathsf{good}}$, a contradiction. This disproves (42) and therefore completes the proof of the trade-off lower bound in (13). $\qquad\square$

# F   Proofs for Section B

With slight abuse of notation, we define function $f(x)$ for any $x \in \mathbb{R}_{\geq 0}$ as

$$f(x) \equiv \sum_{a \in [K]} \frac{\overline{\sigma}_a^2 \vee (x + \overline{\Delta}_a)}{T(x + \overline{\Delta}_a)^2}.$$

Let us consider the following fixed-point equation in $\varphi$

$$f(\varphi) = \sum_{a \in [K]} \frac{\overline{\sigma}_a^2 \vee (\varphi + \overline{\Delta}_a)}{T(\varphi + \overline{\Delta}_a)^2} = 1. \tag{43}$$

The following lemma extends Lemma 1 to the $K$-armed setting.

**Lemma 7.** *It holds that:*

1. *The fixed-point equation (43) admits a unique solution $\varphi^\star \in (1/T, 1)$ for all $T \in \mathbb{N}_+$.*

2. *Assume that there exist some $\delta \in (-1/2, 1/2)$ and $\varphi_\delta$ such that $f(\varphi_\delta) = 1 + \delta$. Then there exists some constant $c = c(K) > 0$ such that*

$$\left|\frac{\varphi_\delta}{\varphi^\star} - 1\right| \leq c|\delta| \cdot \max_{a \in [K]}\left\{1 + \left(\frac{1}{K-1} - \frac{\overline{\sigma}_a^2}{T\overline{\Delta}_a^2}\right)_+^{-1} \wedge \left(\frac{\overline{\sigma}_a^2}{T\overline{\Delta}_a^2} - 1\right)_+^{-1} \wedge \frac{\overline{\sigma}_a}{\sigma_1 \vee T^{-1/2}}\right\}. \tag{44}$$

*Proof.* The proof is almost the same as that of Lemma 1. We include the proof here for the convenience of readers.

(1). First we prove the existence and uniqueness of $\varphi^\star$. Note that $f(\varphi)$ is continuously monotone decreasing on $(0, \infty)$. On the other hand, as

$$f(1/T) = \frac{\sigma_1^2 \vee T^{-1}}{T^{-1}} + \frac{\sigma_2^2 \vee (T^{-1} + \overline{\Delta}_a)}{T(T^{-1} + \overline{\Delta}_a)^2} > \frac{\sigma_1^2 \vee T^{-1}}{T^{-1}} \geq 1$$

and

$$f(1) = \sum_{a \in [K]} \frac{\overline{\sigma}_a^2 \vee (1 + \overline{\Delta}_a)}{T(1 + \overline{\Delta}_a)^2} \leq \frac{K}{T} < 1,$$

there must exist a unique $\varphi^\star \in (1/T, 1)$ such that $f(\varphi^\star) = 1$, proving the claim.

(2). We only show the proof for the case of $\delta > 0$, while the case of $\delta < 0$ can be handled similarly. It follows from the monotonicity that $\varphi_\delta \leq \varphi^\star$. Let $\varphi_\delta/\varphi^\star = 1 - \overline{\delta}$ for some $\overline{\delta} = \overline{\delta}(\delta) \geq 0$. Our aim is to derive an upper bound for $\overline{\delta}$.

It follows from (43) that at least one of the following holds:

$$\textbf{Case 1}: \frac{\overline{\sigma}_1^2 \vee \varphi^\star}{T(\varphi^\star)^2} \geq \frac{1}{K}, \quad \textbf{Case 2}: \exists a \in [K] \setminus \{1\}, \frac{\overline{\sigma}_a^2 \vee (\varphi^\star + \overline{\Delta}_a)}{T(\varphi^\star + \overline{\Delta}_a)^2} \geq \frac{1}{K}.$$

**Case 1**: If $\overline{\sigma}_1^2 = \overline{\sigma}_1^2 \vee \varphi^\star$, we have $\overline{\sigma}_1^2 = \overline{\sigma}_1^2 \vee \varphi_\delta$. Then it holds that

$$\delta = f(\varphi_\delta) - f(\varphi^\star) \geq \frac{\overline{\sigma}_1^2 \vee \varphi_\delta}{T\varphi_\delta^2} - \frac{\overline{\sigma}_1^2 \vee \varphi^\star}{T(\varphi^\star)^2} = \frac{\overline{\sigma}_1^2}{T(\varphi^\star)^2}\left(\frac{1 - (1 - \overline{\delta})^2}{(1 - \overline{\delta})^2}\right) \geq \frac{1}{K}\overline{\delta}.$$

For the case when $\varphi^\star = \overline{\sigma}_1^2 \vee \varphi^\star$, we can compute that

$$\delta = f(\varphi_\delta) - f(\varphi^\star) \geq \frac{\overline{\sigma}_1^2 \vee \varphi_\delta}{T\varphi_\delta^2} - \frac{\overline{\sigma}_1^2 \vee \varphi^\star}{T(\varphi^\star)^2} \geq \frac{1}{T\varphi^\star}\left(1 - \frac{1}{1 - \overline{\delta}}\right) \geq \frac{1}{K}\overline{\delta}.$$

By combining the above results, in Case 1, we can conclude that $\overline{\delta} \leq K\delta$.

**Case 2**: Note that in this case, we have

$$\frac{1}{T\varphi^\star} \leq \frac{1}{T\varphi^\star} \vee \frac{\overline{\sigma}_1^2}{T(\varphi^\star)^2} \leq \frac{\overline{\sigma}_1^2 \vee \varphi^\star}{T(\varphi^\star)^2} \leq \frac{K - 1}{K}. \tag{45}$$

This implies that when $\varphi^\star + \overline{\Delta}_a = \overline{\sigma}_a^2 \vee (\varphi^\star + \overline{\Delta}_a)$,

$$\begin{aligned}
\delta = f(\varphi_\delta) - f(\varphi^\star) &\geq \frac{1}{T((1 - \overline{\delta})\varphi^\star + \overline{\Delta}_a)} - \frac{1}{T(\varphi^\star + \overline{\Delta}_a)} \\
&= \frac{\overline{\delta}\varphi^\star}{T(\varphi^\star + \overline{\Delta}_a)((1 - \overline{\delta})\varphi^\star + \overline{\Delta}_a)} \\
&\geq \frac{\overline{\delta}T\varphi^\star}{T(\varphi^\star + \overline{\Delta}_a)T(\varphi^\star + \overline{\Delta}_a)} \geq \frac{\overline{\delta}}{K(K - 1)}.
\end{aligned}$$

For the case when $\overline{\sigma}_a^2 = \overline{\sigma}_a^2 \vee (\varphi^\star + \overline{\Delta}_a)$, using (45) and the fact that $\varphi^\star \geq T^{-1}$ leads to

$$\frac{\overline{\sigma}_a^2}{T(\varphi^\star + \overline{\Delta}_a)^2} \geq \frac{1}{K} \geq \frac{1}{K - 1} \cdot \frac{\overline{\sigma}_1^2 \vee \varphi^\star}{T(\varphi^\star)^2} \geq \frac{1}{K - 1} \cdot \frac{\overline{\sigma}_1^2 \vee T^{-1}}{T(\varphi^\star)^2},$$

which entails that $\varphi^\star/(\varphi^\star + \overline{\Delta}_a) \geq (\overline{\sigma}_1 \vee T^{-1/2})/(\overline{\sigma}_a\sqrt{K - 1})$. Hence, we have that

$$\begin{aligned}
\delta = f(\varphi_\delta) - f(\varphi^\star) &\geq \frac{\overline{\sigma}_a^2}{T(\varphi_\delta + \overline{\Delta}_a)^2} - \frac{\overline{\sigma}_a^2}{T(\varphi^\star + \overline{\Delta}_a)^2} \tag{46} \\
&= \frac{\overline{\sigma}_a^2(\varphi^\star - \varphi_\delta)}{T(\varphi^\star + \overline{\Delta}_a)^2} \cdot \left(\frac{\varphi_\delta + \varphi^\star + 2\overline{\Delta}_a}{(\varphi_\delta + \overline{\Delta}_a)^2}\right) \geq \frac{\varphi^\star\overline{\delta}}{K(\varphi^\star + \overline{\Delta}_a)} \geq \frac{\overline{\sigma}_1 \vee T^{-1/2}}{K\sqrt{K - 1}\overline{\sigma}_a}\overline{\delta}.
\end{aligned}$$

This gives the bound that corresponds to $\overline{\sigma}_a/(\overline{\sigma}_1 \vee T^{-1/2})$ in (44).

Next, we will derive the bound corresponding to $(\overline{\sigma}_a^2/(T\overline{\Delta}_a^2) - 1)_+^{-1}$ and $(1/(K - 1) - \overline{\sigma}_a^2/(T\overline{\Delta}_a^2))_+^{-1}$ in (44). The analysis will be further separated into two cases: (i) $1/(K - 1) - \overline{\sigma}_a^2/(T\overline{\Delta}_a^2) \geq 0$ and (ii) $1 - \overline{\sigma}_a^2/(T\overline{\Delta}_a^2) < 0$.

For case (i), notice that

$$\frac{\overline{\sigma}_1^2 \vee \varphi^\star}{T(\varphi^\star)^2} = 1 - \sum_{a \geq 2} \frac{\overline{\sigma}_a^2 \vee (\varphi^\star + \overline{\Delta}_a)}{T(\varphi^\star + \overline{\Delta}_a)^2}$$

$$\geq 1 - (K-1) \cdot \max_{a \geq 2} \frac{\overline{\sigma}_a^2 \vee (\varphi^\star + \overline{\Delta}_a)}{T(\varphi^\star + \overline{\Delta}_a)^2}.$$

Let $a_\star \equiv \arg\max_{a \geq 2} (\overline{\sigma}_a^2 \vee (\varphi^\star + \overline{\Delta}_a))/(T(\varphi^\star + \overline{\Delta}_a)^2)$. Then we can show that

$$\frac{\overline{\sigma}_1^2 \vee \varphi^\star}{T(\varphi^\star)^2} \geq (K-1) \cdot \left( \frac{1}{K-1} - \frac{\sigma_{a_\star}^2}{T\Delta_{a_\star,T}^2} \right)$$

Thus, we can arrive at

$$\delta = f(\varphi_\delta) - f(\varphi^\star) \geq \frac{\overline{\delta}}{1-\overline{\delta}} \cdot \frac{\overline{\sigma}_1^2 \vee \varphi^\star}{T(\varphi^\star)^2} \geq (K-1) \cdot \left( \frac{1}{K-1} - \frac{\sigma_{a_\star}^2}{T\Delta_{a_\star,T}^2} \right) \cdot \overline{\delta}$$

For case (ii), let us recall from (46) that

$$\delta \geq \frac{\varphi^\star \overline{\delta}}{2(\varphi^\star + \overline{\Delta}_a)} \geq \frac{\overline{\delta}}{3K} \cdot \mathbf{1}_{\overline{\Delta}_a/\varphi^\star \leq 2} + \frac{\varphi^\star \overline{\delta}}{2(\varphi^\star + \overline{\Delta}_a)} \cdot \mathbf{1}_{\overline{\Delta}_a/\varphi^\star > 2}. \tag{47}$$

Observe that when $\overline{\Delta}_a/\varphi^\star > 2$ (or equivalently $\varphi^\star/\overline{\Delta}_a < 1/2$),

$$\frac{\overline{\sigma}_a^2}{T(\varphi^\star + \overline{\Delta}_a)^2} \geq \frac{\overline{\sigma}_a^2}{T\overline{\Delta}_a^2}\left(1 - 3\frac{\varphi^\star}{\overline{\Delta}_a}\right) \Rightarrow \left(\frac{\overline{\sigma}_a^2}{T\overline{\Delta}_a^2} - 1\right)\frac{1}{\varphi^\star} - \frac{3\overline{\sigma}_a^2}{T\overline{\Delta}_a^3} \leq 0.$$

Using the condition that $1 - \overline{\sigma}_a^2/(T\overline{\Delta}_a^2) < 0$, we can deduce that

$$(\varphi^\star)^{-1}\overline{\Delta}_a \leq \frac{3\overline{\sigma}_a^2}{T\overline{\Delta}_a^2} \cdot \left(\frac{\overline{\sigma}_a^2}{T\overline{\Delta}_a^2} - 1\right)^{-1}.$$

As $\overline{\sigma}_a^2/(\varphi^\star + \overline{\Delta}_a)^2 \leq T$ and $\varphi^\star/\overline{\Delta}_a < 1/2$, we can obtain that $\overline{\sigma}_a^2/(T\overline{\Delta}_a^2) \leq 9/4$, which then leads to $(\varphi^\star)^{-1}\overline{\Delta}_a \leq 27/(2\overline{\sigma}_a^2/(T\overline{\Delta}_a^2) - 2)$. Hence, it follows that

$$\frac{\varphi^\star \overline{\delta}}{2(\varphi^\star + \overline{\Delta}_a)} \cdot \mathbf{1}_{\overline{\Delta}_a/\varphi^\star > 2} \geq \frac{\overline{\delta}}{2 + 27\left(\frac{\overline{\sigma}_a^2}{T\overline{\Delta}_a^2} - 1\right)^{-1}} \cdot \mathbf{1}_{\overline{\Delta}_a/\varphi^\star > 2}.$$

Plugging this back to (47) yields that

$$\delta \geq \frac{\overline{\delta}}{3K + 27\left(\frac{\overline{\sigma}_a^2}{T\overline{\Delta}_a^2} - 1\right)^{-1}}.$$

By combining the above results, in Case 2, we can conclude that

$$\overline{\delta} \leq (3K + 27)\delta \cdot \max_{a \in [K]} \left\{ 1 + \left(\frac{1}{K-1} - \frac{\overline{\sigma}_a^2}{T\overline{\Delta}_a^2}\right)_+^{-1} \wedge \left(\frac{\overline{\sigma}_a^2}{T\overline{\Delta}_a^2} - 1\right)_+^{-1} \wedge \frac{\overline{\sigma}_a}{\overline{\sigma}_1 \vee T^{-1/2}} \right\}.$$

The claim then follows by combining the estimates in Cases 1 and 2 above. □

*Proof of Theorem 4.* The proof of Theorem 4 closely mirrors that of Theorem 1, and we spell out some of the details for the convenience of readers. We start by noticing that most of the derivations in Section D.2 hold for the $K$-armed setting. Specifically, from (27) we have that

$$\sum_{a \in [K]} \frac{\overline{\sigma}_a^2 \vee (\varphi_{1,T} + \overline{\Delta}_a)}{T(\varphi_{1,T} + \overline{\Delta}_a)^2} = 1 + \mathfrak{o}_{\mathbf{P}}(1). \tag{48}$$

Combining (48) above with the stability result in Lemma 7 concludes the proof. □

*Proof of Theorem 5.* See the proof of Theorem 2 in Section E.2. □

## G Auxiliary results

**Lemma 8.** *Assume that $X_1, X_2, \ldots$ are i.i.d. bounded random variables in $[-1, 1]$ with zero mean and variance $\sigma^2$. Let $\bar{X}_t \equiv \sum_{s \in [t]} X_s / t$ be the empirical mean. Then we have that for any $\delta < 1/2$,*

$$\mathbb{P}\left(|\bar{X}_t| < \sigma \sqrt{\frac{12 \log \frac{\log(T/\delta)}{\delta}}{t}} + \frac{64 \log(1/\delta)}{t}, \ \forall 1 \leq t \leq T\right) \geq 1 - \delta.$$

*Proof.* As in previous works, our proof relies on constructing a supermartingale and applying Doob's inequality. Let $\mathcal{F}_t$ be the filtration generated by $X_1, \ldots, X_t$ and $S_t \equiv \sum_{s \in [t]} X_s$. For any $\eta \in [0, 1]$,

$$\mathbb{E}[\exp(\eta S_t)|\mathcal{F}_{t-1}] = \exp(\eta S_{t-1}) \, \mathbb{E}[\exp(\eta X_t)|\mathcal{F}_{t-1}]$$
$$\leq \exp(\eta S_{t-1}) \left(1 + \eta^2 \sigma^2\right) \leq \exp(\eta S_{t-1} + \eta^2 \sigma^2). \tag{49}$$

Here, in the first inequality above, we have used the fact that $\mathbb{E}[\exp(\eta X_t)] \leq 1 + \eta^2 \sigma_a^2$; see, e.g., [17, Lemma 7].

With $\gamma_i \equiv i^{-1}(i+1)^{-1}$ and $\eta_i \in [0, 1]$ to be determined, let us define

$$Z_t \equiv \begin{cases} 1, & \text{if } t = 0, \\ \sum_{i=1}^{\infty} \gamma_i \exp\left(\eta_i S_t - t\eta_i^2 \sigma^2\right), & \text{if } t \geq 1. \end{cases} \tag{50}$$

Then in light of (49), it holds that

$$\mathbb{E}[Z_{t+1}|\mathcal{F}_t] = \sum_{i=1}^{\infty} \gamma_i \left(\mathbb{E}[\exp\left(\eta_i S_{t+1} - t\eta_i^2 \sigma^2\right)|\mathcal{F}_t] \cdot \exp(-\eta_i^2 \sigma^2)\right)$$
$$\leq \sum_{i=1}^{\infty} \gamma_i \exp\left(\eta_i S_t - t\eta_i^2 \sigma^2\right) = Z_t,$$

implying that $\{Z_t\}_{t \geq 0}$ is sequence of supermartingale. Note that by invoking a similar argument as in [23, Proof of Lemma 5.1],

$$\{Z_t \geq 1/\delta\} \supset \left\{\max_i \gamma_i \exp\left(\eta_i S_t - t\eta_i^2 \sigma^2\right) \geq 1/\delta\right\}$$
$$= \left\{\bar{X}_t \geq \min_i \left(\frac{1}{t\eta_i} \log(\frac{1}{\delta\gamma_i}) + \eta_i \sigma^2\right)\right\}.$$

For any fixed $\delta > 0$, set

$$\eta_i = \begin{cases} 2^{-i/2} \sigma_a^{-1} \log^{1/2}\left(\frac{i(i+1)}{\delta}\right), & \text{if } i \geq i_1, \\ 1, & \text{if } i < i_1, \end{cases}$$

where $i_1 \equiv \inf\{i \in \mathbb{Z}_{\geq 1} : 2^{\bar{i}} \geq \sigma_a^{-2}(\log(\bar{i}(\bar{i}+1)) + \log(1/\delta)) \text{ holds for } \forall \, \bar{i} \geq i\}$.

Given these parameters, the analysis is further divided into two cases: $\sigma_a \geq 1/\sqrt{T}$ and $\sigma_a < 1/\sqrt{T}$.

**Case 1:** For $\sigma_a \geq 1/\sqrt{T}$, it holds that

$$\{Z_t \geq 1/\delta\} \supset \left\{\bar{X}_t \geq \min_i \left(\frac{1}{t\eta_i} \log(\frac{1}{\delta\gamma_i}) + \eta_i \sigma^2\right)\right\}$$
$$\supset \left\{\bar{X}_t \geq \min_{i \geq i_1} \left(\sqrt{\frac{t}{2^i}} + \sqrt{\frac{2^i}{t}}\right) \cdot \sigma \frac{\log^{1/2}\left(\frac{i(i+1)}{\delta}\right)}{t^{1/2}}\right\}.$$

For those $t$ satisfying $\lceil \log_2 t \rceil \geq i_1$, we can choose $i = \lceil \log_2 t \rceil$ to obtain

$$\{Z_t \geq 1/\delta\} \supset \left\{\bar{X}_t \geq 4\sigma \frac{\log^{1/2}\left(\frac{2(\log 4t)^2}{\delta}\right)}{t^{1/2}}\right\}.$$

For those $t$ satisfying $\log_2 t \leq i_1 - 1$, it follows from the definition of $i_1$ that $t < 2^{i_1}$ and $2^{i_1-1} \geq \sigma^{-2}(\log(i_1(i_1-1)) + \log(1/\delta))$. Hence, we can deduce that

$$
\begin{aligned}
\min_{i \geq i_1} & \left( \sqrt{\frac{t}{2^i}} + \sqrt{\frac{2^i}{t}} \right) \cdot \sigma \frac{\log^{1/2}\left(\frac{i(i+1)}{\delta}\right)}{t^{1/2}} \\
& \leq \left( 1 + \sqrt{2}\sqrt{\frac{2^{i_1-1}}{t}} \right) \cdot \sigma \frac{\log^{1/2}\left(\frac{i_1(i_1+1)}{\delta}\right)}{t^{1/2}} \\
& \leq \sigma \frac{\log^{1/2}\left(\frac{i_1(i_1+1)}{\delta}\right)}{t^{1/2}} + \sqrt{2\log(i_1(i_1-1)) + 2\log(1/\delta)} \cdot \frac{\log^{1/2}\left(\frac{i_1(i_1+1)}{\delta}\right)}{t}.
\end{aligned}
\tag{51}
$$

Below we will derive an upper bound for $i_1$. Let us consider functions $f(x) = 2^x$ and $g(x) = \sigma^{-2}\log(x(x+1)) + \log(1/\delta)$. It is easy to show that

$$
f'(x) = 2^x \log 2 > 2^{x-1}, \quad g'(x) = \frac{2x+1}{\sigma^2 x(x+1)} < \frac{2T}{x}.
$$

Then we have $f'(x) > g'(x)$ as long as $x > \log(2T) + 1$. On the other hand, for $x_0 = 4\log(1/\delta) + 4\log T + 1$, it follows from $\sigma^{-2} \leq T$ that

$$
f(x_0) = 2(T/\delta)^4 > 2T \log(5\log(2T/\delta)) + \log(1/\delta) > g(x_0),
$$

which entails that $f(x) > g(x)$ holds for all $x > x_0$. Thus, we have $i_1 \leq x_0 = 4\log(T/\delta) + 1$.

Combining this upper bound of $i_1$ with (51), we can obtain that

$$
\{Z_t \geq 1/\delta\} \supset \left\{ \bar{X}_t \geq \sigma \sqrt{\frac{12 \log \frac{\log(T/\delta)}{\delta}}{t}} + \frac{64\log(1/\delta)}{t} \right\}.
$$

**Case 2:** For $\sigma < 1/\sqrt{T}$, we can deduce that

$$
\begin{aligned}
\{Z_t \geq 1/\delta\} & \supset \left\{ \bar{X}_t \geq \min_i \left( \frac{1}{t\eta_i} \log(\frac{1}{\delta\gamma_i}) + \eta_i \sigma^2 \right) \right\} \\
& \supset \left\{ \bar{X}_t \geq \min_{i < i_1} \left( \frac{1}{t\eta_i} \log(\frac{1}{\delta\gamma_i}) + \eta_i \sigma^2 \right) \right\} \\
& \supset \left\{ \bar{X}_t \geq \frac{\log(1/\delta) + \log 2}{t} + \frac{1}{T} \right\} \supset \left\{ \bar{X}_t \geq \frac{3\log(1/\delta)}{t} \right\}.
\end{aligned}
$$

The claim follows. $\qquad\square$

As a corollary of Lemma 8, we have the following results on the concentration of the empirical mean and variance: for any $\delta \geq 0$, let

$$
\mathsf{e}(\delta; T) \equiv \sqrt{48\rho \log T \log \frac{\log(T/\delta)}{\delta}} + 128\log(1/\delta).
$$

**Proposition 7.** *For all $\delta < 1/2$ and $a \in [K]$, it holds that*

$$
\mathbb{P}\left( \left| \bar{X}_{a,t} - \mu_a \right| < \varphi_{a,t} \cdot \mathsf{e}(\delta; T)/2, \ \forall 1 \leq t \leq T \right) \geq 1 - \delta,
\tag{52}
$$

$$
\mathbb{P}\left( \left| \hat{\sigma}_{a,t}^2 - \sigma_a^2 \right| < \varphi_{a,t} \cdot \mathsf{e}(\delta; T), \ \forall 1 \leq t \leq T \right) \geq 1 - 2\delta.
\tag{53}
$$

*Proof.* The claim in (52) follows from a direct application of Lemma 8 with the following simplification for the error bound therein

$$
\sigma_a \sqrt{\frac{12 \log \frac{\log(T/\delta)}{\delta}}{n_{a,t}}} + \frac{64\log(1/\delta)}{n_{a,t}} \leq \left( \bar{\sigma}_a \vee n_{a,t}^{-1/2} \right) \cdot \frac{\mathsf{e}(\delta; T)}{2\sqrt{n_{a,t}}} = \varphi_{a,t} \cdot \frac{\mathsf{e}(\delta; T)}{2\sqrt{n_{a,t}}}.
$$

To prove (53), we first notice that by $|\mu_a - \bar{X}_{a,t}| \leq 1$,

$$|\hat{\sigma}_{a,t}^2 - \sigma_a^2| = \left| \frac{1}{n_{a,t}} \sum_{s \in [n_{a,t}]} \left( (X_{a,s} - \mu_a)^2 - \sigma_a^2 \right) - (\mu_a - \bar{X}_{a,t})^2 \right|$$

$$\leq \left| \frac{1}{n_{a,t}} \sum_{s \in [n_{a,t}]} (X_{a,s} - \mu_a)^2 - \sigma_a^2 \right| + |\mu_a - \bar{X}_{a,t}|.$$

Then we can bound the second term above using (52). The first term above can be analyzed by the same argument as in proving (52), upon noting that $\mathrm{Var}((X_{a,s}-\mu_a)^2-\sigma_a^2) \leq \mathbb{E}((X_{a,s}-\mu_a)^4) \leq \sigma_a^2$. This concludes the proof. $\qquad \square$

**Lemma 9.** *Assume that $\{X_i\}_{i \geq 1} \overset{i.i.d.}{\sim}$ Bernoulli$(1/2)$ and fix any $\mathsf{c} > 0$. Then there exist some $N_0 \in \mathbb{Z}_+$ and constant $c_{anti} = c_{anti}(\mathsf{c}) > 0$ such that for all $N \geq N_0$,*

$$\mathbb{P}\left( \sum_{i \in [n]} X_i > \frac{n}{2} + \sqrt{n}, \ \forall \lfloor N/\mathsf{c} \rfloor \leq n \leq N \right) \geq c_{anti}, \tag{54}$$

$$\mathbb{P}\left( \sum_{i \in [n]} X_i < \frac{n}{2} - \sqrt{n}, \ \forall \lfloor N/\mathsf{c} \rfloor \leq n \leq N \right) \geq c_{anti}. \tag{55}$$

*Proof.* Our proof relies on Donsker's Theorem, which states the convergence (in distribution) of the random walk to the continuous-time Brownian motion; see, e.g., [6, Section 14]. More precisely, for $\{X_i\}_{i \geq 1} \overset{i.i.d.}{\sim}$ Bernoulli$(1/2)$, Donsker's Theorem states that for

$$\Psi_N(t) \equiv \frac{2}{\sqrt{N}} \left( \sum_{i \in [\lfloor Nt \rfloor]} X_i - \frac{Nt}{2} \right),$$

it holds uniformly in $t \in [0,1]$ that $\lim_{N \to \infty} \Psi_N(t) \overset{d}{=} B(t)$, where $B(t)$ represents the standard Brownian motion. Equivalently, for any $\varepsilon > 0$, there exists some $N_0 \in \mathbb{Z}_+$ such that for all $N \geq N_0$,

$$\sup_{x \in \mathbb{R}} \left| \mathbb{P}\left( \inf_{t \in [0,1]} \Psi_N(t) \geq x \right) - \mathbb{P}\left( \inf_{t \in [0,1]} B(t) \geq x \right) \right| \leq \varepsilon.$$

Using the fact that $\inf_{\lfloor N/\mathsf{c} \rfloor \leq n \leq N} 2(\sum_{i \in [n]} X_i - n/2)/\sqrt{N} \geq \inf_{t \in [1/\mathsf{c},1]} \Psi_N(t)$, we can deduce that for any $x \in \mathbb{R}$,

$$\mathbb{P}\left( \sum_{i \in [n]} X_i \geq \frac{n}{2} + x\sqrt{N}, \ \forall \lfloor N/\mathsf{c} \rfloor \leq n \leq N \right)$$

$$\geq \mathbb{P}\left( \frac{1}{2} \inf_{t \in [1/\mathsf{c},1]} \Psi_N(t) \geq x \right) \geq \mathbb{P}\left( \inf_{t \in [1/\mathsf{c},1]} B(t) \geq 2x \right) - \varepsilon. \tag{56}$$

As $B(t + \mathsf{c}^{-1}) \overset{d}{=} Z + \bar{B}(t)$ for another standard Brownian motion $\bar{B}(t)$ and $Z \sim \mathcal{N}(0, \mathsf{c}^{-1})$ independent from $\bar{B}(t)$, we can further bound the right-hand side (RHS) of the inequality in the above display as follows. Denote by $\Phi$ the cumulative distribution function (CDF) of the standard Gaussian distribution. Then we can show that

$$\text{RHS of (56)} = \mathbb{P}\left( \inf_{t \in [0, 1-\mathsf{c}^{-1}]} \bar{B}(t) \geq 2x - Z \right) - \varepsilon \overset{(i)}{=} \mathbb{P}\left( \sup_{t \in [0, 1-\mathsf{c}^{-1}]} \bar{B}(t) \leq Z - 2x \right) - \varepsilon$$

$$\overset{(ii)}{\geq} \mathbb{P}\left( \sup_{t \in [0, 1-\mathsf{c}^{-1}]} \bar{B}(t) \leq x \right) \cdot \mathbb{P}\left( Z \geq 3x \right) - \varepsilon$$

$$\overset{(iii)}{=} \left( 2\Phi\left( \frac{x}{\sqrt{1 - \mathsf{c}^{-1}}} \right) - 1 \right) \cdot \left( 1 - \Phi\left( 3\sqrt{\mathsf{c}} x \right) \right) - \varepsilon.$$

Here, in step (i) above we have used the symmetry property of the Brownian motion, in step (ii) above we have used the independence between $\bar{B}(t)$ and $Z$, and in step (iii) above we have used the reflection principle.

Now choosing

$$x = 1, \quad c_{anti} = \frac{1}{2}\left( 2\Phi\left( \frac{x}{\sqrt{1 - \mathsf{c}^{-1}}} \right) - 1 \right) \cdot \left( 1 - \Phi\left( 3\sqrt{\mathsf{c}} x \right) \right), \quad \varepsilon = c_{anti}$$

finishes the proof for (54). The proof for (55) is nearly the same due to the symmetric property of the Brownian motion, so will be omitted here. $\qquad \square$

# H   Numerical simulations

This section summarizes the settings of all numerical experiments reported in the main text and specifies the reward distributions. Unless otherwise specified, we consider two-armed bandit instances and take arm 1 to be the optimal arm.

**Common numerical settings for generating reward distributions.** In all experiments, we use a $\text{Beta}(\alpha, \beta)$ distribution to generate rewards. Given a desired mean $\mu$ and variance $\sigma^2$, and subject to the boundedness constraint on $[0, 1]$, we set

$$\alpha = \mu\left(\frac{\mu(1-\mu)}{\sigma^2} - 1\right), \qquad \beta = (1-\mu)\left(\frac{\mu(1-\mu)}{\sigma^2} - 1\right).$$

Note that an implicit constraint on $(\mu, \sigma^2)$ is $\sigma^2 \le \mu(1-\mu)$; this constraint is satisfied by all reward distributions used below.

**Figure 1: Distributions of $n_{1,T}$.** In both experiments that plot histograms of arm-pull counts, we set the time horizon to $T = 50,000$ and the number of repetitions to $R = 5,000$. The exploration hyperparameter is $\rho = 2$ for both UCB-V and UCB. The means and variances are set to $\mu_1 = \mu_2 = \frac{1}{2}$ and $\sigma_1 = \sigma_2 = \frac{1}{4}$ in Figure 1(a), and to $\sigma_1 = 0$, $\sigma_2 = \frac{1}{4}$, and $\Delta = \sigma_2 \sqrt{(\log T)/T}$ in Figure 1(b).

**Figure 2: Regret and phase transition of optimal-arm pulls.** For panel (a), we set the times of repetition as 10, exploration hyper-parameter $\rho = 2$. We vary $\sigma_1 \in \{T^{-1/2}, T^{-1/4}, 1\}$ while keeping $\sigma_2 = T^{-1/4}$ and $\mu_1 = \frac{1}{2}, \Delta_2 = T^{-1/2}, \mu_2 = \mu_1 + \Delta_2$ fixed across curves. We plot the realized regret $\text{Reg}(T)$ versus $T$ and observe the trend toward $\mathcal{O}\big((\sigma_2^2/\sigma_1)\sqrt{T}\big)$ as $\sigma_1$ increases. For panel (b), we set $T = 1,000,000$, and repetition time $R = 30$, $\rho = 2$, and fix $\sigma_1 = 0$ and $\sigma_2 = \frac{1}{4}$. We sweep $\Lambda_T = \sigma_2 \sqrt{\rho \log T}/(\sqrt{T} \Delta_2)$ by varying $\Delta_2$; for each value we plot the median and the 30% quantile of $n_{1,T}$ for UCB and UCB-V. The red dotted curve shows the numerical solution $n^\star_{1,T}$ from (6), revealing a transition from sublinear to linear $n_{1,T}$ around $\Lambda_T \approx 1$.

# I   Further discussion on inference with UCB-V

The asymptotic normality of the $Z$-statistic in (8) is established via the martingale CLT. Due to the adaptive nature of data collection in the bandit algorithms, traditional inference techniques based on independent and identically distributed (i.i.d.) data cannot be applied directly. However, for stable arms, let us consider the filtration $\mathcal{F}_s$ generated by $\{X_1, \ldots, X_s\}$. The sequence $\left\{\frac{\mathbf{1}\{a_s=a\}(X_s-\mu_a)}{\sigma_a \sqrt{n^\star_{a,T}}}\right\}_{s \in [T]}$ forms a martingale difference sequence. Moreover, the Lindeberg condition is satisfied, and thus it holds that $\sum_{s \in [T]} \mathbb{E}\left(\frac{\mathbf{1}\{a_s=a\}(X_s-\mu_a)^2}{\sigma_a^2 n^\star_{a,T}} \mid \mathcal{F}_{s-1}\right) \xrightarrow{p} 1$, which allows for the application of the martingale CLT to establish the asymptotic normality of the $Z$-statistic. For a detailed derivation, see [23, Section 2.1].

To illustrate the implication of our results on the reward inference, we conduct a simulation study on the $Z$-statistic for UCB and UCB-V using the setting from Example 2, as shown in Figure 4. In Figure 4a, the empirical distributions of the $Z$-statistic for both UCB and UCB-V approximate the standard Gaussian, matching the theoretical predictions of our result in Example 2, where UCB-V is stable in this regime. In Figure 4b, the empirical distribution of the $Z$-statistic for UCB-V shows a noticeable bias compared to UCB. This suggests that the previously mentioned martingale CLT result no longer holds for UCB-V when $\Lambda_T = 1$. In the subsequent section, we show that the underlying reason for this deviation from the CLT is the *instability* of UCB-V under condition $\Lambda_T = 1$.

# J   Discussion on the modification of UCB-V

In our presentation of Algorithm 1, there are several differences compared to those proposed in [4]. In [4], the UCB reward for each action $a$ is set as

$$B_{a,t} \equiv \bar{X}_{a,t} + \widehat{\sigma}_{a,t}\sqrt{\frac{2\zeta \log t}{n_{a,t}}} + c\frac{3\zeta \log t}{n_{a,t}},$$

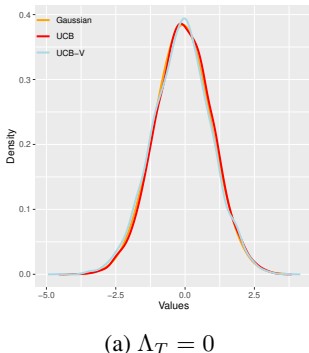
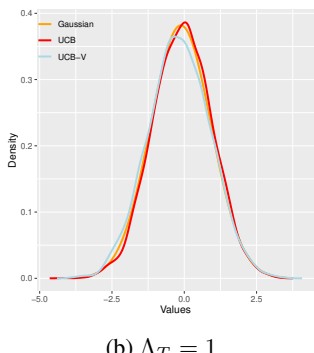

(a) $\Lambda_T = 0$            (b) $\Lambda_T = 1$

Figure 4: The empirical distributions of the $Z$-statistic for the sub-optimal arm for UCB and UCB-V with $\sigma_1 = 0, \sigma_2 = 1/4$ under different $\Lambda_T$, both with $T = 50,000$ over $2,000$ repetitions.

where $c, \zeta > 0$ are tunable constants. Specifically, their regret guarantee is established with $c, \zeta$ large enough. In our statement, to simplify the notation in our asymptotic analysis, we choose $c$ and $\zeta$ such that for some $\rho > 0$, it holds that $2\zeta = 3c\zeta = \rho$, replacing $\log t$ with $\log T$ and modifying

$$\widehat{\sigma}_{a,t}\sqrt{\frac{\rho \log T}{n_{a,t}}} + \frac{\rho \log T}{n_{a,t}} \quad \text{to} \quad \left(\widehat{\sigma}_{a,t} \vee \sqrt{\frac{\rho \log T}{n_{a,t}}}\right) \cdot \sqrt{\frac{\rho \log T}{n_{a,t}}}.$$

We make such modification for notational simplicity and facilitating comparison with results for the canonical UCB [21, 23]. By setting $\widehat{\sigma}_{a,t} = 1$ instead of estimating it via the sample variance in the algorithm's input, our analysis shows that the asymptotic behavior of the modified algorithm becomes equivalent to that of the canonical UCB.

