# OpenReview forum: "Precise Asymptotics and Refined Regret of Variance-Aware UCB"
_NeurIPS.cc/2025/Conference — NeurIPS 2025 spotlight_

### Official Review · Reviewer_1t6H · 2025-06-08

**Clarity:** 4
**Significance:** 3
**Originality:** 3
**Rating:** 6
**Confidence:** 5

**Summary:**

This paper studies the UCB-V algorithm for stochatic bandit, a variant of the UCB algorithm when the standard deviation needs to be estimated. In particular, this paper studies the asymptotic arm pulls and how they depend on the gap between the suboptimality gap and the reward variances. The result shows a phase transition, as the arm pulls can be linear or sublinear based on the these parameters.

**Questions:**

The author focuses on a specific implementation of UCB-V. In the class of UCB algorithms, there is one variant that doesn't use $\log T$ in the numerator in the square root but $\log t$. See Chapter 8 of the book by Lattimore and Szepesvar. This variant is less conservative and has better empirical performance especially when $T$ is large. I wonder if (1) UCB-V has such a variant and (2) how easy/challenging is it to adapt the current analysis to the variant.

**Ethical Concerns:**

["NO or VERY MINOR ethics concerns only"]

**Quality:**

4

**Strengths And Weaknesses:**

Strengths:
- This paper is a theoretical paper with heavy notation. But the authors are able to present the result and explain the intuition really well. Overall I enjoy reading it and learned a lot from the results and analyses.
- The results are highly nontrivial and comprehensive. The asymptotic rate of the arm pulls depend on a lot of parameters, and the authors are able to analyze the dependence very thoroughly.
- The discussion of the literature is detailed and the readers can see which analyses are inspired by which paper and how the results differ.

Weakness: NA

---

> ### Author Rebuttal · Authors · 2025-07-30
>
> Thank you for your interest in our work and for reviewing our manuscript! We greatly appreciate your feedback and address your comment below.
>
>
>
> **Question: Modifying current analysis to the $\log t$ variant**
>
> Thank you for raising this insightful question!
>
> (1) In [4],  UCB-V is presented in the general form
>     \begin{align*}
>          \mathrm{UCB}(a,t) = \bar{X}\_{a,t} +  \widehat \sigma\_{a,t} \cdot \sqrt{\frac{\mathcal{E}\_t}{n\_{a,t}}}
>     +\frac{\mathcal{E}\_t}{n\_{a,t}},
>     \end{align*}
> where $\mathcal{E}_t$ is any non-decreasing function of $t$. Setting $\mathcal{E}_t = \log t$ recovers the ``$\log t$'' variant you mentioned.
>
> (2) We believe extending our current analysis to the ``$\log t$'' variant is **not** straightforward. Technically, it requires at least two major changes:
>
> - **An improved Bernstein‑type non‑asymptotic LIL result.**
>
> In our Lemma 8, the Bernstein-type non-asymptotic LIL is stated as: with probability at least $1-\delta$,
>     \begin{align*}
>         \big| \bar{X}_t\big| \lesssim \sigma\sqrt{\frac{\log\frac{\log(T/\delta)}{\delta}}{t}} + \frac{\log(1/\delta)}{t},  \forall 1 \leq t \leq T.
>     \end{align*}
>     To handle the ``$\log t$'' variant, we may need to replace the fixed threshold $\log\frac{\log(T/\delta)}{\delta}$ by a time-varying $\log\frac{\log(t/\delta)}{\delta}$ inside the square root, i.e.,
>     \begin{align*}
>         \big| \bar{X}_t\big| \lesssim \sigma\sqrt{\frac{\log\frac{\log(t/\delta)}{\delta}}{t}} + \frac{\log(1/\delta)}{t},  \forall 1 \leq t \leq T.
>     \end{align*}
>
> Although obtaining such an improvement might require a more refined martingale-based analysis, we are hopeful that a stronger bound could be achievable.
>
>
>
> - **A revised UCB‑V analysis framework.**
>
>  Even with the refined LIL concentration inequality we mentioned in point 1, under the ``$\log t$'' variant our current analysis framework breaks down. Concretely, one obtains
>
> \begin{align*}
>         1 + \mathfrak{o}\_{\mathbf{P}}(1) \geq \frac{\Delta_a / \log T_a+ \varphi\_{1,T} }{ \varphi_{a,T}} ,\quad \frac{\Delta_a / \log T_1+ \varphi\_{1,T} }{ \varphi\_{a,T}} \geq 1 -\mathfrak{o}\_{\mathbf{P}}(1),
> \end{align*}
>
> where $\log T_1$ and $\log T_a$ now differ because of the time-varying bonus. Such mismatch prevents us from collapsing to a single fixed point equation.
>
> A similar limitation appears in the non-variance UCB analysis in [23], which also studies the \``$\log T$'' version of the UCB algorithm, and this may be a fundamental gap in using such a framework to conduct analysis. It is worth noting that [21] has successfully analyzed the behaviour of non-variance UCB in the \``$\log t$'' variant, but with a different, dedicated analysis and with results in a less complete characterization (as discussed below Theorem 3.1 of [23]). We tend to believe that this subtle limitation of the framework used by [23] and our work is a fundamental price paid for providing more complete characterization and simplifying the analysis of [21]. Given the less conservative nature and strong empirical performance of this  \``$\log t$'' variant algorithm, exploring it further could be a worthwhile avenue for future work.

---

> > ### Comment · Reviewer_1t6H · 2025-08-03
> >
> > Thanks for your explanation on the technical challenge of this extension. I don't have further questions.

---

> > > ### Author Response · Authors · 2025-08-04
> > >
> > > Thank you for your feedback!

---

### Official Review · Reviewer_nRgh · 2025-06-14

**Clarity:** 3
**Significance:** 3
**Originality:** 3
**Rating:** 5
**Confidence:** 4

**Summary:**

This paper studies the asymptotic behavior of a variance-aware MAB algorithm called UCB-V in the small gap regime, which reveals significant differences compared to the classic UCB algorithm in the variance-agnostic setting. First, when the deviation of the best arm $\sigma_1$ and the suboptimal arm $\sigma_2$ are of the same order, the asymptotic arm pulling fraction is proportional to the variance of each arm, instead of an equal split in UCB. Second, when $\sigma_1$ is significantly smaller than the sub-optimal arm $\sigma_2$, the arm pulling fraction may be unstable, depending on the ratio between $\sigma_2$ and the reward gap $\Delta$, where a phase transition phenomenon is also revealed. Third, a refined regret bound involving the variance of the best arm $\sigma_1$ is developed, which is strictly better than the current known result, with a matching lower bound and numerical validation.

**Questions:**

The example to illustrate the instability of the arm pulling number may not be convincing enough. Since the best arm has variance $0$, and the algorithm knows it,  we do not need to construct confidence intervals for the best arm. A reasonable and natural modification to conduct in this setting is to only pull arm $1$ only once, instead of following the UCB-V algorithm itself. Then, there would not be an instability case since we only pull arm 2. Could you construct a bandit instance with positive variance to show the instability in Proposition 2?

**Ethical Concerns:**

["NO or VERY MINOR ethics concerns only"]

**Final Justification:**

Technically solid and strong paper, which, in my opinion, should be accepted. It reveals the fundamental behavior of an important bandit algorithm: UCB-V. The paper has quite a strong impact on advancing our understanding of bandit and RL problems, and is of importance to the ML community.

The two-armed setting may be restrictive, but extending the mathematical tools in this paper and previous studies to the multi-armed case is generally a very difficult problem. Therefore, the reasons to accept this paper greatly outweigh the reasons to reject.

**Limitations:**

Yes

**Paper Formatting Concerns:**

No concerns on paper formatting is found.

**Quality:**

4

**Strengths And Weaknesses:**

Strengths

1. A solid and comprehensive theoretical investigation is conducted on the two-armed MAB setting, which reveals a significant behavior difference between UCB-V and UCB in asymptotic regimes.

2. The paper characterizes the arm-pulling fraction based on the study of the asymptotic equation. An unstable regime is revealed when the best arm has low variance, and the gap is in the order of $\sigma_2\sqrt{\theta \log T/ T}$ especially when the constant $\theta$ is close to the chosen exploration constant $\rho$. By a perturbation analysis of the stable equation, a divergent behavior and a phase transition phenomenon are discovered, which justifies the additional effect of variance estimation.

3. The influence of variances of the best arm and sub-optimal arms on the regret performance is also captured via Theorem 2, where a bound involving the best arm's variance is provided. This result (with extension to the multi-armed setting) captures the specific role of the best arm's variance in addition to sub-optimal arms, which is a nice complement to the current worst-case bandit analysis literature. A matching lower bound is also provided to characterize its fundamentality.

Weaknesses:

1. Overall, this paper could improve in several areas regarding presentation and elaboration: (1) I would suggest elaborating on the precise asymptotic behavior for UCB-V in the introduction and using a table to summarize the findings in different regimes; (2) The authors could use a separate section to discuss their numerical results and how it is consistent with theoretical findings. Those results are merely mentioned but not fully discussed in the current version; (3) the addition notation paragraph in section 2 could be expanded so that each new notation, such as the stability function $\phi$, is given a physical meaning to conceptually prepare for the stability analysis; and (4) the paper is missing a conclusion section.

2. Although extensions to K-armed settings are also discussed for the regret bound, it is hard to generalize the stability results to general K-armed cases. Therefore, the insight into the unstable phase transition behavior of UCB-V may be limited since the two-armed model is a very special case.

---

> ### Author Rebuttal · Authors · 2025-07-30
>
> Thanks for your interest in our work and for reviewing our manuscript. We greatly appreciate your feedback and address your comment below.
>
>
> **1. Writing suggestions**
>
> Thank you for these valuable suggestions! We apologize for the roughness of some discussion sections and for omitting a conclusion due to page limitations. In the revised version, we will:
>
> i) Expand the introduction to state our UCB‑V asymptotics explicitly and include a compact table summarizing our stability results and regret rates across different regimes.
>
> ii) Add a new section that consolidates all simulation results, detailing each experimental setup and explaining how each experiment aligns with our theoretical findings.
>
> iii) In Section 2’s Additional Notation paragraph, explicitly define the stability function
> $$ \varphi\_{a,t} = \frac{\bar{\sigma}\_a \vee n_{a,t}^{-1/2}}{\sqrt{n_{a,t}}}, $$
> and clarify that these two components correspond to the two error terms in Bernstein's inequality—so that the UCB‑V bonus for arm $a$ at time $t$ is taken as the larger of the variance‑based and boundedness‑based corrections.
>
>
> iv) Add a Concluding section that summarizes our main contributions, acknowledges limitations, and outlines directions for future work.
>
>
> **2. Limitations in $K$ armed setting.**
>
> We agree that our stability condition (14) may not be optimal for $K>2$, and that a full phase‑transition characterization is still lacking. The main technical challenge in extending our theory to the $K>2$ case lies in the increased number of parameters and their complex interactions. We therefore leave a tighter analysis of the $K>2$ case as an important direction for future work.
>
>
> **3. Instability example beyond zero‑variance instances**
>
> We chose $\sigma_1 = 0$ solely for simplicity of exposition. In fact, the instability analysis in Proposition 2 extends directly to settings where the optimal arm’s variance satisfies $\sigma_1 = \mathcal{O}(1 /\sqrt{T})$, since the UCB‑V bonus term for the optimal arm is the same as in the $\sigma_1 = 0$ case. We will generalize Proposition 2 accordingly in the revised manuscript.
>
> On the other hand, we  would like to note that constructing an instance with $\sigma_1 = T^{-\alpha}$ for some $\alpha \in (0,1/2)$ is more challenging and requires some new ideas, as the asymptotic equation from Example 2 no longer applies.  We will acknowledge such limitation in the revised version.

---

> > ### Comment · Reviewer_nRgh · 2025-08-01
> >
> > Thanks to the authors for their response to my questions, and I'm happy the authors are willing to take my suggestions to improve clarity.
> >
> > I want to explain to the authors why I'm not giving a 6. The reason is that the technical tools, to my knowledge, may not be meaningful in multi-armed problems. It somewhat diminishes its importance in practical bandit applications with hundreds or thousands of arms with small gaps, since our understanding there is not fully clear.
> >
> > Nevertheless, the paper is of theoretical interest itself and has implications in some applications, such as A/B tests. In my point of view, this paper stands out among all papers I reviewed this year, as a submission of high quality which no doubt exceeds the standard of Neurips. I would be very happy to recommend it for acceptance with full confidence.

---

> > > ### Author Response · Authors · 2025-08-02
> > >
> > > Thank you! We sincerely appreciate your valuable suggestions and feedback.

---

### Official Review · Reviewer_vmvf · 2025-06-26

**Clarity:** 3
**Significance:** 2
**Originality:** 3
**Rating:** 5
**Confidence:** 4

**Summary:**

This paper studies the statistic performance of the UCB-V algorithm in multi-armed bandits. The authors first give a stable condition, and show that under this condition, the number of pulls on every arm is predictable. Then they show that if the stable condition does not hold, the number of pulls on arms could vary a lot in random. Based on this analysis, the authors also provide refined regret upper bounds for the UCB-V algorithm: if the optimal arm's variance is small, the proposed regret upper bound matches existing ones; however, if the optimal arm's variance is large, the proposed regret upper bound can becomes smaller than existing ones. They also use a lower bound to show the tightness of their refined regret upper bound.

**Questions:**

I do not have many questions. Please see "Weaknesses" part above.

**Ethical Concerns:**

["NO or VERY MINOR ethics concerns only"]

**Final Justification:**

The answer is clear, I do not have further questions.

I believe the paper's technical contributions are clear and meaningful to the bandit literature. Therefore, I recommend accepting the paper.

**Limitations:**

Yes.

**Quality:**

3

**Strengths And Weaknesses:**

Strengths:

- The writing is clear, and the results are easy to understand.
- The previous regret upper bound is improved (although not a large one).
- The stability analysis may also give insights for the analysis in other cases, as variance-adaptive algorithms are widely applied.


Weaknesses:

- My main concern is that the most technically challenging case analyzed in this paper—namely when the gap $\Delta = \Theta(1/\sqrt{T})$—is rather rare in practice. Since the paper focuses heavily on this regime, the practical relevance of the results may be limited.

- Some parts of the exposition could be clarified. For example, in lines 85–86, the meanings of the notations "$\gtrsim$" and "$\lesssim$" are not very clear. Additionally, in line 166, the use of $\delta$ to denote a perturbation is potentially confusing, as $\delta$ is already used as the confidence level in Proposition 1.

---

> ### Author Rebuttal · Authors · 2025-07-30
>
> Thank you for your interest in our work and for reviewing our manuscript! We greatly appreciate your feedback and address your comments below.
>
> **Question 1: $\Delta \asymp 1/\sqrt{T}$ setting of interest**
>
> We agree with the reviewer that $\Delta = \Theta(1/\sqrt{T})$ represents a delicate “edge case” in practical applications. However, this regime plays a crucial conceptual and methodological role: by parameterizing $\Delta \asymp \frac{\theta}{\sqrt{T}}$ and letting $\theta\to0$ or $\theta\to+\infty$, one interpolates smoothly between the small-gap and large-gap regimes. In addition to this interpolation perspective, we borrow examples and explanations from [21] to illustrate how small-gap phenomena arise in practice:
>
> - **A/B testing and clinical trials.**
>   With large sample budgets $T$, the minimum detectable effect at fixed significance and power satisfies
>   $T \propto \Delta^{-2}
>     \Longrightarrow
>     \Delta = \mathcal{O}\bigl(1/\sqrt{T}\bigr).$
>   Hence, web experiments and vaccine dosage trials naturally operate in this small-gap regime.
>
> - **Online allocation of homogeneous tasks.**
>   Many platforms (ride‑hailing, crowdsourcing) route statistically indistinguishable tasks to agents, requiring fairness on each sample path. Here, true performance differences shrink to $\mathcal{O}(1/\sqrt{T})$, so early random fluctuations can lead to stark imbalances in task allocation.
>
> Under such small‑gap regime, prior works [21, 23] showed that UCB algorithms can maintain low regret and balanced allocations. By contrast, our analysis reveals that incorporating variance estimates yields lower regret but the allocation depends on variances and may not be balanced. Such trade-off highlights a fundamental tension between efficiency and fairness: variance‑aware policies achieve better performance by pulling high-variance arms more often, yet this comes at the cost of an unbalanced allocation, which may be undesirable in applications.
>
> **Question 2: Notational confusions**
>
> Thank you for your careful reading! We have corrected the noted notational issues in the revised version:
>
> 1. In line 85, we will update the inequality to
> $$
> \mathbb{P}\bigl(n_{1,T} \lesssim \sqrt{T}\log T\bigr)
>      \wedge
>      \mathbb{P}\bigl(n_{2,T} \gtrsim T/(\sigma_2\sqrt{\log T})\bigr)
>      \gtrsim 1.
> $$
> Moreover, in the Notation section we will add: we write $a \gtrsim b$ (resp. $a \lesssim b$) to mean $a \geq C b$ (resp. $a \leq Cb $) for some absolute constant $C > 0$.
>
> 2. We will replace the perturbation symbol $\delta$ in line 166 with
> $\zeta$ to avoid confusion.

---

> > ### Comment · Reviewer_vmvf · 2025-08-09
> > **Reply**
> >
> > Thanks for your answers. I do not have further questions.

---

### Official Review · Reviewer_XTVW · 2025-07-01

**Clarity:** 3
**Significance:** 3
**Originality:** 3
**Rating:** 4
**Confidence:** 1

**Summary:**

This paper studies the behaviour of the Upper Confidence Bound-Variance (UCB-V) algorithm for the Multi-Armed Bandit (MAB) problems. More precisely, it provides an asymptotic characterization of the arm-pulling rates for UCB-V, extending recent results for the canonical UCB. The author establishes that UCB-V can achieve a more refined regret bound.

**Questions:**

The extension to the K-armed setting is mentioned but not deeply explored. Could the authors provide more intuition or empirical results for this case?

**Ethical Concerns:**

["NO or VERY MINOR ethics concerns only"]

**Final Justification:**

The authors' clarification addressed my concerns. I don't have further questions. Thus I maintain my score.

**Limitations:**

As I mentioned in the weaknesses and questions, this paper only considers the two-armed setting. This setup currently has no practical application because very few real-world scenarios meet the two-armed setting criteria. However, I believe it is essential to consider the two-armed setting before addressing the multi-armed setting.

**Quality:**

3

**Strengths And Weaknesses:**

Strengths:

  - This paper provides a detailed analysis of precise asymptotic behaviour for UCB-V, like the stability of arm-pulling numbers.

- This paper derives non-asymptotic bounds for the arm pulling numbers in the high probability regime, which shows high probability bounds and confidence regions for arm pulling numbers.

- The authors include a refined regret for variance-aware decision making.

Weaknesses:

- This paper only focuses on the two-armed setting.

---

> ### Author Rebuttal · Authors · 2025-07-30
>
> Thank you for your interest in our work and for reviewing our manuscript. We greatly appreciate your feedback and address your comments below.
>
> **Question: Extension to the $K$‑armed setting**
>
> As mentioned in our last section and Appendix B, we can derive analogous results in the $K$‑armed setting—including the asymptotic equation (15), the sufficient condition for stability (14), and the improved regret bound in Theorem 5—which all reduce to the two‑armed results presented in the main text. We would like to discuss several additional points and calculations:
>
> _1. Asymptotic arm‑pulling characterization._ To provide some intuition for (15), we can generalize Example 1 from the two‑armed setting: assume all variances $\sigma_a=\Omega(1)$ and the gaps $\Delta_a$ are proportional to $\theta_a\sqrt{\log(T)/T}$—the “moderate‑gap regime” of [21]. In this regime, condition (14) always holds, and (15) simplifies to
>
> $$
> \sum_{a=1}^K \frac{\sigma_a^2}{T\bigl(\sigma_1/\sqrt{n_{1}^\star} + \theta_a/\sqrt{T}\bigr)^2} = 1,
> \qquad
> n_a^\star = \frac{\sigma_a^2}{T\bigl(\sigma_1/\sqrt{n_{1}^\star} + \theta_a/\sqrt{T}\bigr)^2}.
> $$
>
> As $\theta_a\to0$, arm‑$a$’s normalized pulling rate ($n_a^\star/T$) converges to
>
> $$
> \frac{\sigma_a^2}{\sum_{j=1}^K \sigma_j^2}.
> $$
>
> That is, when the gaps shrink, each arm’s asymptotic pulling rate becomes proportional to its variance. This generalizes the intuition from Example 1 and the homogeneous setting in [23]. We will include this as an illustrative example in the revised manuscript.
>
> Besides this case (i.e., $\sigma_a=\Omega(1)$ for all $a$), we note that, unlike the two‑armed setting, our stability condition (14) may not be tight for general $K$: it is intrinsically harder to construct a counterexample as in Section 3.3 due to the increased number of parameters. In (14), we require an arm‑wise boundedness condition to guarantee convergence to the solution of (15), but we conjecture that it might be relaxed to a single, global condition over all arms.
>
> _2. Regret in the K‑armed setting._  When
> $$
> \sqrt{\sum_{a}\sigma_a^2}\lesssim \sum_{a\ge2}\frac{\sigma_a^2}{\sigma_1},
> $$
> our regret bound in Theorem 5 reduces to
> $$
> \widetilde O\bigl(\sqrt{\sum_{a\ge2}\sigma_a^2 T}\bigr),
> $$
> matching previous variance‑aware results. However, in the regime of small $\sigma_a$ and large $\sigma_1$, our bound improves to
> $$
> \sum_{a\ge2}\frac{\sigma_a^2}{\sigma_1}\sqrt{T},
> $$
>
> which is near‑optimal in the two‑armed case (Theorem 3). Interestingly, for general $K$, this rate scales linearly in $K$, unlike the previous bound which scales as $\sqrt{K}$; we conjecture that such linear dependence may be unavoidable for UCB‑V, as supported by our empirical results in the Table below (if plotted, the curve of final cumulative regret versus $K$ exhibits an approximately linear trend). Establishing matching lower bounds remains a valuable direction for future work.
>
> **Table: Regret for UCB‑V under different $K$.**
>
> _Simulation parameters:_ rewards are drawn from the beta distributions with $(\mu_1,\sigma_1)=(0.5,0.4)$ and $(\mu_2, \sigma_2) = \ldots = (\mu_K, \sigma_K) = (0.1, 0.01)$; each simulation is run for  $T = 10,000$ rounds.
> |     $K$    |   2   |    5   |   10   |    20   |    30   |    40   |    50   |
> | :--------: | :---: | :----: | :----: | :-----: | :-----: | :-----: | :-----: |
> | **Regret** | 9.192 | 36.768 | 82.816 | 174.880 | 267.984 | 359.520 | 450.480 |
>
>
>
> **Limitation: Practical relevance of the two‑armed setting**
>
> We agree that a full treatment of the $K$-armed case is both important and challenging. Nonetheless, the two-armed setting itself actually admits several concrete applications. For example,  A/B testing in web experiments or marketing campaigns naturally reduces to two arms (control vs.  treatment).
>     The analysis in the two-armed setting also provides useful insights for the eventual $K$-armed extension. We hope this explanation is helpful.

---

> > ### Comment · Reviewer_XTVW · 2025-08-05
> > **Response**
> >
> > Thanks for the clarification. I don't have further questions.

---

### Decision · Program_Chairs · 2025-09-17

**Decision:**

Accept (spotlight)

**Comment:**

This paper presents a theoretical analysis of the Upper Confidence Bound-Variance (UCB-V) algorithm for multi-armed bandit problems. The reviewers unanimously praised the paper for its technical depth, clarity, and originality, recognizing it as a solid and comprehensive investigation into the behavior of an important variance-aware algorithm. While some reviewers initially raised concerns about the primary focus on the two-armed setting and its practical relevance, the authors' rebuttal effectively addressed these points by providing strong justifications for the importance of this setting in applications like A/B testing and by offering clear intuition and empirical support for the extension to the more general K-armed case. The authors also agreed to incorporate several presentation improvements suggested by the reviewers, further strengthening the manuscript. All reviewers were satisfied with the authors' responses, leading to a consensus of acceptance.